# Modifying electron injection kinetics for selective photoreduction of nitroarenes into cyclic and asymmetric azo compounds

Yang Yang[1], Xu Jing[1✉], Jing Zhang[1], Fengyu Yang[1] & Chunying Duan [1✉]

Modifying the reactivity of substrates by encapsulation is essential for microenvironment catalysts. Herein, we report an alternative strategy that modifies the entry behaviour of reactants into the microenvironment and substrate inclusion thermodynamics related to the capsule to control the electron injection kinetics and the selectivity of products from the nitroarenes photoreduction. The strategy includes the orchestration of capsule openings to control the electron injection kinetics of electron donors, and the capsule's pocket to encapsulate more than one nitroarene molecules, facilitating a condensation reaction between the in situ formed azanol and nitroso species to produce azo product. The conceptual microenvironment catalyst endows selective conversion of asymmetric azo products from different nitroarenes, wherein, the estimated diameter and inclusion Gibbs free energy of substrates are used to control and predict the selectivity of products. Inhibition experiments confirm a typical enzymatic conversion, paving a new avenue for rational design of photocatalysts toward green chemistry.

[1] State Key Laboratory of Fine Chemicals, Zhang Dayu College of Chemistry, Dalian University of Technology, Dalian 116024, China.
✉email: xjing@dlut.edu.cn; cyduan@dlut.edu.cn

Catalytic synthetic methods inspired by natural prototypes that react under ambient conditions using green energy sources are one of the important fields in synthetic chemistry[1–3]. To echo the laudable catalytic properties and underpin the reactivity and selectivity that are not observed in bulk solution, chemists employ microenvironment catalysts that mimic the pocket and loaded catalytic active sites of natural enzymes[4–6]. Enzymes, however, change not only the reactivity of substrates but also the properties of other essential factors to enhance catalytic activities[7–9]; such as reactants entry, substrate inclusion[10], and entropic generation[11,12]. Alternatively, modifying the electron donation kinetics to optimize the reaction pathways by controlling the diffusion behaviour of reactants through the openings of hosts remains unexplored, especially for selective photocatalytic conversion.

Recent advances have demonstrated that dye-loaded redox-active capsules are capable of enhancing the stability of charge separation states[13], via the fast pseudo-intramolecular electron transfer from the excited state of dyes to the metal ions, for light-driven hydrogen evolution and hydrogenation reactions using the in situ formed green H-source[14,15]. We envisioned that fine tuning the entry behavior of electron donors into the hosts would balance the kinetics between electron injection and substrate reduction, generating unique selectivity different from the reported abiotic manifolds[16]. One of the important catalytically relevant but still sluggish conversions is selective reduction of nitroarenes into azo products[17–20], due to the inherent complexity of the nitro reduction processes[21–24] and the distinctive application of azo compounds in the areas of liquid crystals, photochemical switches and sensitive shuttles (Fig. 1a)[25–28]. We

proposed that with the capsule was large enough to encapsulate more than one substrate molecules, the slower electron injection kinetics from the electron donors to the dye-loaded capsule would retard the electron transfer to recover the photosensitizer, dominating a stepwise reduction of nitroarenes. Such that in situ formed azanol and nitroso intermediates possibly condensed together to form the corresponding azo products within the capsule before they were further reduced or squeezed out of the capsule[29–31].

Noted that the commercial azo dyestuffs were commonly prepared via the well-known diazotization method using diazonium salts as reaction intermediates[32,33] and recent advances demonstrated several one-pot procedures for reductive condensation of nitroarenes into aromatic azo compounds[34–38]. However, reductive coupling of two different nitroarenes into one asymmetric azo compound remains challenging, due to the difficult to preclude competing reactions between the azanol and nitroso species with the same aromatic groups from the condensation reactions between the azanol and nitroso species having different aromatic groups. Should the dye-loaded capsule be orchestrated to selectively encapsulate a desired azanol and nitroso pair, asymmetric azo compounds would be formed from the mixture of nitroarenes (Fig. 1b)[39,40].

Herein, a cobalt-based molecular octahedron with robust openings to control reactants entrance and electron injection kinetics into the inner space is assembled for the one-pot photoreduction of nitroarenes into cyclic and asymmetric azo products. Considering the fast-dynamic equilibrium of the clathrates[41–43] in solution, the entry kinetics and the inclusion Gibbs free energy of the substrates are utilized first time to

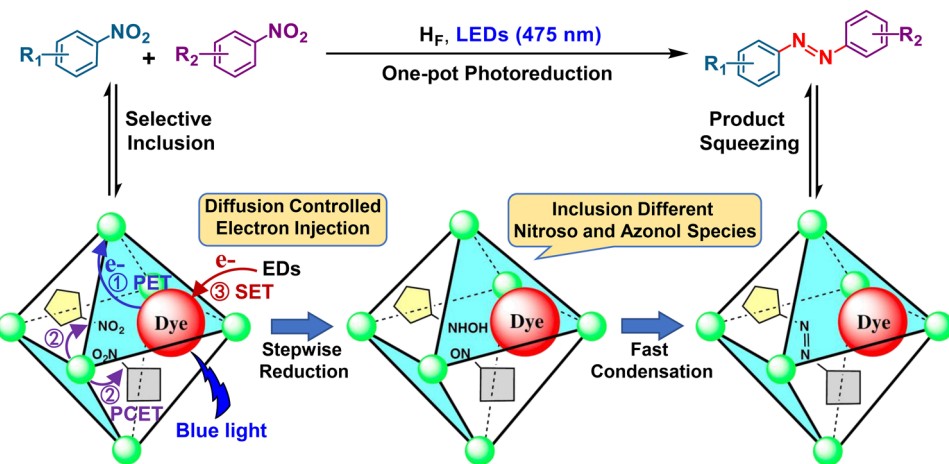

**Fig. 1 Schematic of typical reduction of nitroarenes and one-pot photoreduction of nitroarenes in dye-loaded catalyst $H_F$ system. a** Typical reduction pathways of nitroarenes for amino and azo products. **b** The one-pot photoreduction of nitroarenes into asymmetric azo products using the dye-loaded catalyst $H_F$, showing the proposed electron transfer sequence: initialed from the photoinduced electron transfer (PET) from the excited state of dye to the metal center, then the proton coupled electron transfer from the reduced metal center to the substrate (PCET), followed by the single electron transfer (SET) from the electron donor to the oxidized dye, recovering the organic dye for another reduction.

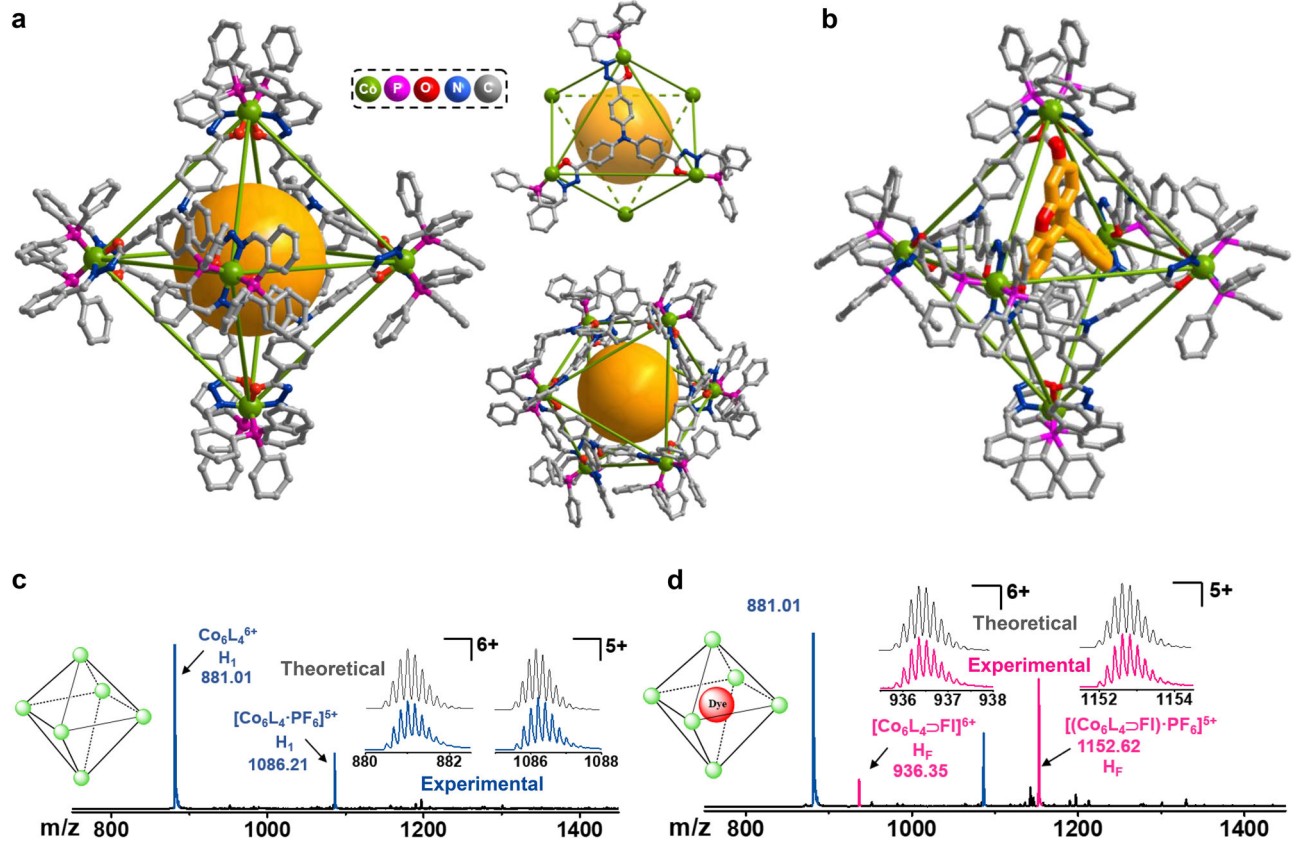

**Fig. 2 Characterization of metal-organic capsule $H_1$ and $H_F$. a** The structure of the octahedral cage $Co_6L_4^{6+}$ ($H_1$) showing the large cavity volume and robust openings. **b** The simulation structure of cage $Co_6L_4^{6+}$ with fluorescein encapsulated showing the relative positions. **c** ESI–MS spectra of metal-organic capsule $H_1$ (1.0 mM) in a $CH_3CN$ solution. **d** ESI–MS spectra of $H_1$ (1.0 mM) with fluorescein (1.0 mM) in a $CH_3CN$ solution. The inserts show the theoretical and experimental isotopic patterns of the peaks mentioned.

control and predict the selectivity of the photoreduction into an asymmetric azo compound over the symmetric azo products. As azanol species always exhibit richer hydrogen bonding sites than that of the corresponding nitroso species, the favorable condensation is achieved between the in situ formed azanol partner from the nitro substrate reduction with a larger retention kinetics and in situ formed nitroso partner with larger entry kinetics into the capsule, when the capsule is large enough to encapsulate two nitroarene molecules. To explore the catalytic mechanism details and apply the innovative enzymatic manifold to prepare azo-containing fine chemicals, nitro substrates with different size of aromatic fragments and different interaction anchors are investigated to obtain asymmetric azo compounds and pharmacologically important cyclic diazocines. Inhibition competing experiments are also displayed to confirm that photocatalytic conversions occur inside the cavity through a typical enzymatic fashion, innovating an interesting avenue for rational design and optimization of photocatalysts toward green chemistry.

## Results and discussion

**Synthesis and characterization of the metal-organic capsule.** The metal-organic capsule $H_1$, $(Co_6L_4)\cdot(PF_6)_6$, was synthesized by refluxing cobalt salts and ligand $H_3L$, tri(2-(diphenylphosphanyl) benzylidene)benzene-4,4',4"-tricarbohydrazinetriphenyl-amine, in a methanol solution with a yield of 70%. The positively charged capsule, $Co_6L_4^{6+}$, is defined by an alternating arrangement of four deprotonated ligands ($L^{3-}$) on its eight triangular faces with a cavity of approximately 1350 Å$^3$ and four robust, triangle-shaped openings having a Co…Co separation about 14.5 Å (Fig. 2a). Each $Co^{III}$ ion is chelated in an octahedral geometry by two widely

delocalized NOP chelators from two different ligands that are positioned in a *mer* configuration, equipping $Co_6L_4^{6+}$ with a robust coordination geometry and high stability. Deprotonated amide groups were located within the positively charged capsule, providing static, coordinative and functional properties to the cage-like capsule, which are important for the recognition and activation of suitable substrates[44,45]. Extensive electron delocalization over the ligand skeleton was found and was expected to facilitate a conventional electron and proton transfer pathway between the inner and outer sphere, making the capsule competent for the holoenzyme mimetic of the catalytic domain in natural enzymes[46,47].

The ESI–MS spectrum of $H_1$ (1.0 mM) in a $CH_3CN$ solution exhibited two intense peaks at $m/z = 881.0157$ and $1086.2099$, which were assignable to species of $Co_6L_4^{6+}$ and $[(Co_6L_4)\cdot(PF_6)]^{5+}$, respectively, through a comparison with the simulation results based on natural isotopic abundances, revealing the presence of trivalent cobalt ions and the high stability of $Co_6L_4^{6+}$ in solution (Fig. 2c). Upon the addition of 1.0 mM fluorescein (donated as Fl) to the $H_1$ (1.0 mM) solution, new peaks corresponding to $[(Co_6L_4)\cdot Fl]^{6+}$ and $[(Co_6L_4)\cdot Fl\cdot(PF_6)]^{5+}$ at $m/z = 936.3547$ and $1152.6194$, respectively, appeared in the ESI–MS spectrum (Fig. 2d), suggesting that one fluorescein molecule was encapsulated in the cavity of $Co_6L_4^{6+}$ to form a 1:1 host-dye species (donated as $H_F$).

$^1H$ NMR spectra of $H_1$ (1.0 mM) in a $CD_3CN$ solution displayed a single set of ligand-related signals corresponding to a highly symmetrical capsule (Supplementary Fig. 15). The $^1H$ NMR titration experiment of $H_1$ (1.0 mM) solution upon addition of Fl (1.0 mM) at 273 K exhibited downfield shifts (0.35 ppm) corresponding to OH protons, along with the upfield

shifts corresponding to aromatic protons in Fl, indicating the encapsulation of fluorescein in the π-electron-rich pocket of the octahedron (Supplementary Fig. 16)[13]. DOSY spectrum of the mixture containing 1.0 mM $H_1$ and 1.0 mM Fl showed that all signals of Fl and $H_1$ had the same diffusion coefficient ($D = 5.19 \times 10^{-10}\,\text{m}^2\text{s}^{-1}$), which also proved the encapsulation between $H_1$ and Fl (Supplementary Fig. 22). Docking calculations by theoretical optimization of the clathrate structure of $H_F$ (Fig. 2b) demonstrated that the organic dye Fl molecule is stuck at one of the openings of $H_1$ and it occupies only a little interior cavity, enabling the further encapsulation of substrates in the interior cavity.

$H_1$ (0.1 mM) gives rise to quasi-reversible $Co^{II}/Co^{I}$ and $Co^{III}/Co^{II}$ couples at −1.31 and −0.55 V (vs. Ag/AgCl), respectively, in a $CH_3CN$ solution (Supplementary Fig. 7). These potentials are in good agreement with the reported cobalt-based metal-organic capsules, wherein the $Co^{II}/Co^{I}$ couple falls well in the range of proton reduction[48,49]. $H_1$ (10.0 μM) quenches the emission of fluorescein (10.0 μM) in a $CH_3CN$ solution (excitation at 475 nm) via pseudo-intramolecular electron transfer (Supplementary Fig. 8)[50,51]. Non-linear fitting of the titration profile gives an associate constant ($K_{a1}$) of $1.79 \pm 0.15 \times 10^5 M^{-1}$ for the 1:1 host-guest clathrate $H_1 \supset$ Fl (donated as $H_F$) between $H_1$ and fluorescein, which is poised to produce hydrogen or H-source for hydrogenation reactions under visible light irradiation.[52] Isothermal titration calorimetry (Supplementary Fig. 38) of the $H_1$ (0.1 mM) solution upon addition of fluorescein confirmed the 1:1 host-dye behavior with a dissociation constant ($K_{d1}$) measuring as $15.7 \pm 1.1$ μM, showing considerable affinity of the clathrate $H_F$ between $H_1$ and fluorescein.

**Size-dependent photoreduction of nitroarenes**. Our photo-catalytic investigation began with the reduction of nitrobenzene to produce aniline using $Na_2S$ as an electron donor[53,54]. Visible light irradiation (475 nm LED) of a $CH_3CN$/$H_2O$ solution (4:1 in volume, 5.0 mL) containing $H_F$ (0.50 μmol $H_1$ and 0.5 μmol fluorescein) and $Na_2S$ (0.60 mmol) resulted in maximum hydrogen generation at pH = 10.0 (Supplementary Fig. 28). Under the standard conditions, the loading of 0.05 mmol nitro-benzene to the aforementioned reaction mixture quenched hydrogen evolution but led to a complete conversion of nitro-benzene into aniline within 30 min. Minimal aniline could be found in the absence of any one of the components. Control experiments using cobalt salt, the ligand itself or the mixture of cobalt salt and the ligand as catalyst, respectively, to replace the metal-organic capsule $H_1$ only produced minimal aniline, even in the presence of identical concentration of Fl. Meanwhile, the use of mononuclear complex $M_1$ (3.0 μmol) (Supplementary Fig. 6) with identical cobalt concentration and Fl concentration could produce aniline but with a yield of 32% at the standard condition. These results indicated the superiority of the dye-loaded photo-catalyst $H_F$ for the reduction of nitrobenzene in the presence of $Na_2S$ as electron donors (Supplementary Table 4).

The initial rate of the reaction exhibits a first-order dependence on the concentrations of the catalyst $H_F$ and the electron donor $Na_2S$, whereas a saturation behavior[55] was observed depending on the concentration of nitrobenzene (Supplementary Fig. 30). Increasing the molecular size from nitrobenzene to 1-nitroanphthalene, 9-nitroanthracene and 1-nitropyrene decreases the initial rate of catalytic reduction (Fig. 3a, Supplementary Figs. 31–33). A positive correlation (Fig. 3b) was recognized between the initial rate of the conversion and the estimated diameter (denoted as d) of substrates (shown as $\pi d^2/4$), which are calculated from the $^1H$ DOSY NMR spectra based on the Stokes–Einstein equation[56,57]. In the case of 2,4,6-triphenyl-nitrobenzene, which has a molecular size that is larger than the opening of $Co_6L_4^{6+}$, was used, no amine product

was detected under the standard condition[58]. When nitrobenzene (0.025 mmol) was added to the reaction mixture containing $H_F$ (0.50 μmol), $Na_2S$ (0.60 mmol) and 0.025 mmol 1-nitroanphtha-lene, 9-nitroanthracene or 1-nitropyrene, respectively (Supplementary Fig. 29), the conversion of aniline in each of the competing reaction increased with the increasing molecular size of the competing substrates (Fig. 3b). The size-dependent results[55,59] revealed that photoreduction probably proceeded inside the capsule via an enzymatic fashion.

Having established the optimal conditions and the size-dependent kinetics of the catalytic process, we next set out to study the influence of the molecular size of electron donors on the catalytic conversion using the middle-sized substrate 9-nitroanthracene. We recognized that irradiating a $CH_3CN$/$H_2O$ solution (4:1 in volume, 5.0 mL) containing 9-nitroanthracene (0.05 mmol), $H_F$ (0.50 μmol) and an electron donor (0.60 mmol), which was N, N'-dimethylaniline, triethylamine, 1-phenylpyrrolidine, N, N'-diethylaniline or tripropy-lamine, respectively, leading to the formation of both aminoanthra-cene and azoanthracene. Increasing the molecular size of electron donors significantly decreased the conversion of the amino product (Fig. 4a), and led to a small but significant decrease in the initial conversion rate for azo compound (Fig. 4b), whereas the ratio between the yields of the azo and amino products exhibited a positive correlation with the estimated diameters but was not dependent on the redox potentials of the electron donors (Fig. 5b).

Upon the addition of the same proportion of the electron donors according to the photocatalytic reaction to the same $H_F$ solution, significant luminescence quenching was observed. We proposed that the interaction between the electron donor and the fluorescein Fl in the microenvironment $H_F$ was essential to affect the luminescence quenching and further control the electron donating kinetics of the photoreduction. We recognized that N, N'-dimethylaniline was the strongest quencher, tripropylamine was the weakest one (Supplementary Fig. 35a), obeying the same tendency to that of the yield ratio between the azo and amino product. We deduced that the conversion rates of the products are mainly controlled by the electron injection from electron donors, but the selectivity of azo compounds over others is mainly controlled by the kinetics difference between the electron injection and condensation reaction in the microenvironment.

As a larger molecular size of electron donors always results in slower entry kinetics or electron injection kinetics into the microenvironment of $H_F$ through the openings, a smaller estimated diameter with faster electron injection kinetics of the electron donor should allow the rapid and continuous reduction of the substrate and intermediates, yielding the final amino product before they are squeezed out of the micro-environment of $H_F$. Alternatively, the larger estimated diameter and slower electron injection kinetics of the electron donor primarily led to a stepwise reduction of nitroarenes into nitroso or azanol intermediates in the microenvironment of $H_F$, yielding the azo compound before they were further reduced to the amine product or squeezed out of the capsule.

Control experiments performed by loading 0.10 mmol electron donors into the aforementioned solution containing 9-nitroanthracene (0.05 mmol) and $H_F$ (0.50 μmol) led to $2e^- + 2H^+$ reduction, giving the intermediate 9-nitrosoanthracene under the standard conditions. Meanwhile, the conversion of 9-nitrosoanthracene increased with the increasing estimated dia-meter of electron donors (Fig. 4c)[60]. These results confirm our assumption that the larger estimated diameter or slower electron injection kinetics of the electron donor lead to a stepwise reduction of the nitroarenes, providing the possibility to condense the in situ formed nitroso or azanol species into azo products.

To this end, we evaluated $H_F$ as an efficient photocatalyst for the synthesis of azo compounds from the reduction of nitroarenes

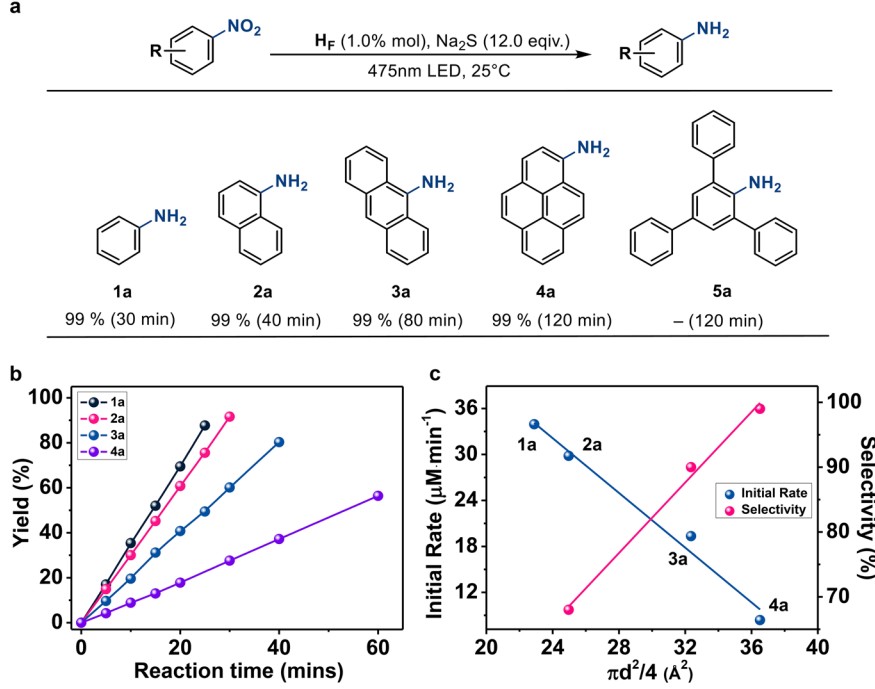

**Fig. 3 Photoreduction of nitroarenes into aminoarenes. a** Photoreduction of nitroarenes into aminoarenes under standard conditions: $H_F$ (0.50 µmol), Na$_2$S (0.60 mmol) and substrate (0.05 mmol) at 25 °C in 5.0 mL CH$_3$CN/ H$_2$O (4:1 in volume) with 475 nm LED; the isolated yields were measured by $^1$H NMR. **b** Kinetics of the aminoarenes conversion with varied substrates under standard conditions. **c** Initial rates and selectivity of aminobenzene over the competing amino products in the reaction mixture containing 0.025 mmol nitrobenzene and competing nitroarene under standard conditions as a function of the estimated diameter (d) of substrates (shown as πd$^2$/4).

under standard conditions: $H_F$ (0.50 µmol), substrate (0.05 mmol) and tripropylamine (0.60 mmol) in a CH$_3$CN/H$_2$O solution (4:1 in volume, 5.0 mL) at pH = 10.0. Excellent conversion and selectivity of the mentioned azo compounds was achieved under 475 nm LED light irradiation for 6 h at room temperature. No azo product was found in the absence of any one of the components, or without the light. Control experiments using cobalt salt, the ligand, the mixture of cobalt salt and the ligand or mononuclear complex $M_1$ to replace $H_1$ as catalysts only produced minimal amino product at the standard conditions even with the presence of identical concentration of Fl. We infer that the dye-loaded photocatalyst $H_F$ is essential for the formation of azo products (Supplementary Table 4).

The initial rate of the reaction indicated a first-order dependence on the concentrations of the catalyst $H_F$ and the electron donor tripropylamine, but exhibited a saturated behavior on the concentration of substrate. Increasing the molecular size by changing the substrate from nitrobenzene to 1-nitroanphthalene and 9-nitroanthracene, respectively, did not impact the initial rate of photocatalytic reduction (Fig. 5a), revealing that the electron injection, not the condensation reaction is the rate-determining step. No azo compound was formed from the reduction of 1-nitropyrene under the standard conditions, as isothermal titration calorimetry (Supplementary Fig. 40) and emission titration (Supplementary Fig. 12) of the $H_F$ solution upon the addition of 1-nitropyrene resulted in an inclusion number of 1.0 (Supplementary Fig. 13). This 1:1 host-guest inclusion behavior thus precludes the possibility of condensing the nitroso and azanol species in one capsule and further supports the photoreductive condensation of nitroarenes occurred inside the cavity in a typical enzymatic fashion. Meanwhile, the only one reduction product of 1-aminopyrene allowed us to explore the potential factor influencing the kinetics of electron injection with different electron donors. A positive

correlation (Supplementary Fig. 35b) between the initial rate of the conversion and the estimated diameter of electron donors demonstrated that the entrance kinetics of electron donors into the microenvironment dominated the electron injection kinetics and the photoreduction conversion rate, confirming that the photocatalytic conversion probably proceeded inside the microenvironment of the capsule.

Recently, a series of metal-loaded catalysts were used to promote the selective reduction of nitroarenes into azo products[61,62], nevertheless, our microenvironment approach represents the first example of an enzymatic protocol that utilizes the entry behavior of electron donors to manipulate the kinetics and selectivity for the preparation of azo products from the reductive condensation of nitroarenes. Different from modifying the reactivity of the substrate and intermediates, our procedure utilizes electron donors with large estimated diameter to retard the electron injection kinetics for the reduction of the in situ formed azanol species into the amino product, echoing the remarkable catalytic properties and underpinning the selectivity that is not observed in bulk solutions. We recognized a positive relationship of the yield ratio between the azo and amino products with the estimated diameter (shown as πd$^2$/4) of electron donors for the reduction of nitrobenzene and 9-nitroanthracene, respectively (Fig. 5b). The relationship followed almost the same trendency with the nitrobenzene and 9-nitroanthracene, confirming that the entrance kinetics of the electron donors into the microenvironment dominated the selectivity for the azo products over the amino products of the photoreductive condensation reactions.

**Inhibition competing conversion.** To further confirm that photocatalytic conversion occurred inside the cavity in a typical enzymatic fashion, inhibition experiments were carried out. Adding up to 7.5 µmol adenosine triphosphate (ATP)[63,64], a nonreactive

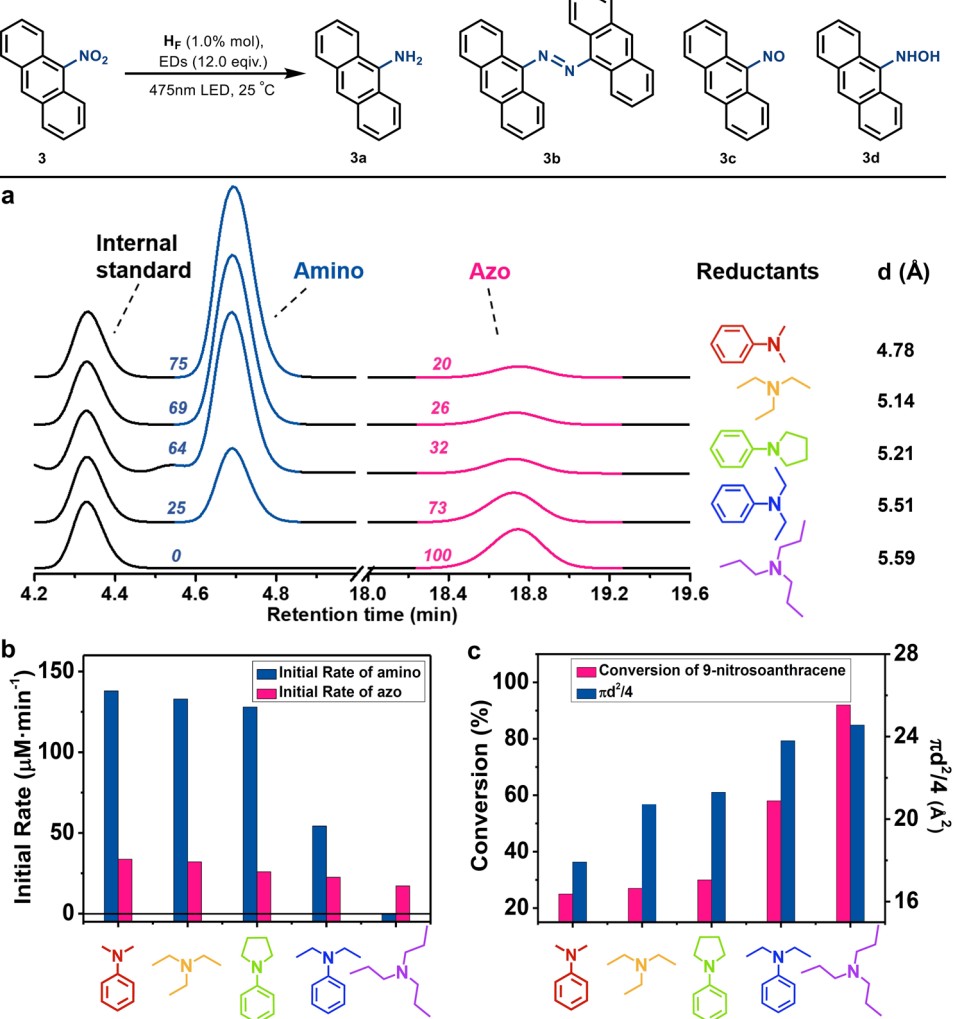

**Fig. 4 Photoreduction of 9-nitroanthracene with different electron donors. a** The conversion of azo and amino products from the reduction of 9-nitroanthracene with various electron donors under standard conditions. **b** The initial conversion rates of the azo and amino product from the reduction of 9-nitroanthracene with various electron donors under standard conditions. **c** The conversion of 9-nitrosoanthracene from the reduction of 9-nitroanthracene with various electron donors as a function of the estimated diameter of substrates (shown as $\pi d^2/4$) of the electron donors.

species (Supplementary Fig. 14), to the reaction mixture containing $H_F$ (0.50 μmol), 9-nitroanthracene (0.05 mmol) and tripropylamine (0.60 mmol) caused a significant decrease in the conversion of azo compounds under the standard conditions (Fig. 6c). We also recognized that the initial rate of the photoreduction of 9-nitroanthracene decreased with the increasing concentrations of ATP added to the reaction mixture (Fig. 6d). The enzymatic reduction of nitroarenes into azo compounds thus proceeded in the microenvironment of the capsule, wherein the dye-loaded host $H_F$ could be defined as a stable microenvironment photocatalyst.

Interestingly, the solution of $H_1$ (20.0 μM) in $CH_3CN$ exhibited an emission band at 510 nm when excited at 365 nm[65]. Upon the addition of 9-nitroanthracene (0.75 mM), the emission was quenched significantly (Supplementary Fig. 11). Hill plot[66] fitting and Job plot[67] fitting of the titration profile suggested 1:2 host-substrate inclusion behavior with the associated constant calculated as $1.69 \pm 0.05 \times 10^7$ $M^{-2}$. The NOESY spectrum of a $CD_3CN$ solution containing $H_1$ (0.10 mM) and 9-nitroanthracene (0.20 mM) shows obvious H−H interactions between the central phenyl ring of the 9-nitroanthracene and the CH=N fragment of $H_1$ and between the phenyl rings from one substrate and the phenyl rings from the other substrate (Supplementary Fig. 24). The interactions enforced the closed

proximity between the 9-nitroanthracene and the host and between the two 9-nitroanthracene molecules, which were beneficial for the encapsulation and stabilization of the clathrate.

Notably, the $CH_3CN$ solution containing $H_1$ (20.0 μM) and fluorescein (20.0 μM) exhibited a ligand-based emission band at 510 nm when excited at 365 nm but did not exhibit a fluorescein-based band, even excited at 475 nm, as $H_1$ strongly quenched the emission of Fl. The addition of 9-nitroanthracene (0.75 mM) to the aforementioned solution quenched the emission strongly of the capsule (Supplementary Fig. 11), but was not able to trigger the fluorescein emission, when excited at 475 nm. These results demonstrate the addition of substrate did not squeeze of the fluorescein out of the capsule.

Further addition of ATP to this solution led to a direct emission recovery of $H_1$ with excitation at 365 nm, but could not trigger the emission of fluorescein when excited at 475 nm (Fig. 6b). As the addition of ATP (0.30 mM) to the $CH_3CN$ solution of $H_1$ (20.0 μM) caused significant emission enhancement (Fig. 6a), we deduce that the substitution of substrate molecules, not the substitution of fluorescein from the capsule, by ATP molecules contributes to competitive inhibition. Isothermal titration calorimetry of the $H_F$ solution containing $H_1$ (0.10 mM)

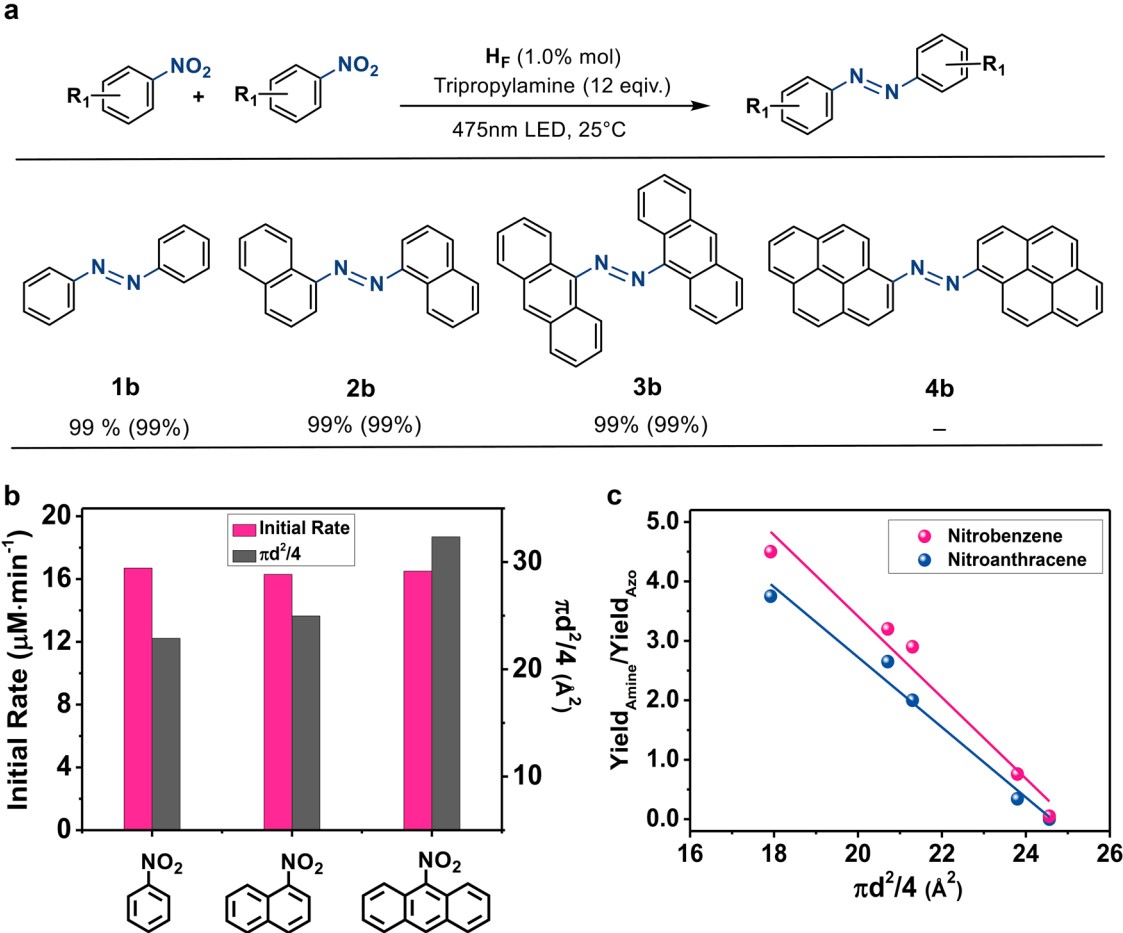

**Fig. 5 Photoreduction of nitroarenes into azo products. a** Photoreduction of nitroarenes into azo products under standard conditions: 1.0 mol% $H_F$, substrates (0.05 mmol) and tripropylamine (0.60 mmol) at 25 °C in 5 mL $CH_3CN/H_2O$ (4:1 in volume) under 475 nm LED irradiation for 6 h. The isolated yields were measured by $^1H$ NMR with the selectivity in parentheses. **b** The initial rate of azo conversion as a function of the estimated diameter of different substrates under standard conditions. **c** The relationship of the ratio between the yield of amino and the yield of azo products as a function of the estimated diameter (shown as $\pi d^2/4$) of electron donors for the reduction of nitrobenzene and 9-nitroanthracene.

and fluorescein (0.10 mM) upon the addition of 9-nitroanthracene revealed the formation of a 1:2 host: guest clathrate between $H_F$ and 9-nitroanthracene, yielding a disassociation constant $K_{d2}$ of $0.76 ± 0.02$ μM (Supplementary Fig. 39). These results revealed the possibility of coexistence of the dye and substrate in the capsule, wherein two 9-nitroanthracene molecules included in one capsule are important for further condensation between the in situ formed nitroso and azanol species within the inner space of the photocatalyst.

To further explain the photocatalytic mechanism, 4-nitrophenol was taken as a model substrate to obtain the corresponding azo compound 4,4'-dihydroxyazobenzene, as the hydroxyl group always worked as an anchoring site to ensure the host-guest interactions being strong enough for easily detection. Isothermal titration calorimetry of the $H_F$ solution upon the addition of 4-nitrophenol and its reduction intermediates revealed that almost all the species exhibited the 1:2 host-guest inclusion behaviour, except the azo product which has two aromatic groups. We also recognize that the azanol species exhibited the largest Gibbs free energy change with $H_F$ over others (Supplementary Table 6). And the catalytic results showed that no azanol products were found from the stepwise reduction of nitroarenes and that the initial rates of the photocatalytic azo conversions were independent by the variation of substrates.

ESI–MS results of the $H_F$ solution upon addition of an equimolar amount of 4-nitrophenol (S), which exhibited the species of $[H_F \cdot 2 S \cdot PF_6]^{5+}$ at $m/z = 1208.2942$, suggested a 1:2 binding ratio between $H_F$ and S that coincides with the isothermal titration calorimetry results (Supplementary Fig. 5). The $^1H$ NMR titration experiment of $H_1$ (1.0 mM) solution upon addition of 4-nitrophenol (10.0 mM) at 273 K exhibited downfield shifts (0.28 ppm) corresponding to OH protons (Supplementary Fig. 17). With addition of 1.0 mM Fl and 10.0 mM 4-nitrophenol in 1.0 mM $H_1$ $CD_3CN$ solution, the downfield shifts of OH proton signals in both Fl and 4-nitrophenol were observed simultaneously, which also matched well with the results in titration process separately and further provided evidences of the co-encapsulation between $H_F$ and 4-nitrophenol (Supplementary Fig. 18). Product inhibition experiments by adding 4,4'-dihydroxyazobenzene up to 0.025 mmol into the reaction mixture containing $H_F$ (0.50 μmol), 4-nitrophenol (0.05 mmol) and tripropylamine (0.60 mmol) only caused slight decrease in the conversion of azo compounds under the standard conditions. Meanwhile the initial rate of the photoreduction of 4-nitrophenol remained unchanged with the increasing concentrations of 4,4'-dihydroxyazobenzene at the beginning of the reaction (Supplementary Fig. 36), which indicated that the substrate could probably squeeze out the product and keep the reaction proceeding continuously. These results support our postulation and innovate a

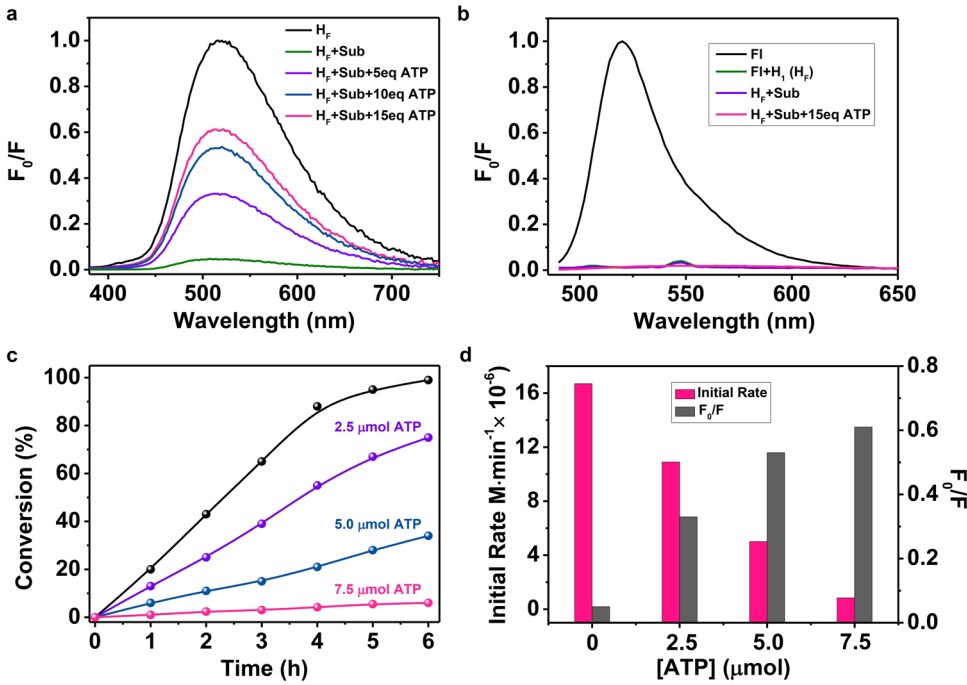

**Fig. 6 The inhibition competing experiments of ATP. a, b** Emission spectra of a $CH_3CN$ solution containing $H_F$ (20.0 μM) and 9-nitroanthracene (0.75 mM) upon the addition of 0.1 mM, 0.2 mM and 0.3 mM ATP when excited at 365 nm or at 475 nm. **c** Conversion of 9-azoanthracene under standard conditions upon addition of 2.5 μmol, 5.0 μmol and 7.5 μmol ATP. **d** The initial conversion rate of the azo product from 9-nitroanthracene under standard conditions and the recovery of capsule emission upon the addition of ATP at different concentrations.

new avenue to prepare azo products under environmentally friendly conditions using easy-to-handle nitroarene feedstocks.

**Photoreduction of nitroarenes into cyclic and asymmetric azo compounds.** To investigate the catalytic mechanism details and apply the innovative enzymatic manifold to prepare azo-containing fine chemicals, substrates with two nitro groups were used to obtain pharmacologically important cyclic diazocines[68]. Loading 1.0 mol% $H_F$ yielded an acyclic azobenzene from 1-nitro-2-[2-(2-nitro-phenoxy)ethoxy]benzene (0.05 mmol), wherein the diazene was embedded in an eleven-membered ring with a yield of 62% upon irradiation for 10 h with tripropylamine (1.20 mmol) as an electron donor. Substrates with different flexible chains were tested in the photocatalytic system, and the rings of the cyclic diazocine products ranged from eight- to eleven-membered (Fig. 7). Compared to the reported synthetic approach of cyclic diazocines[69], the one-pot strategy does not require modification of the catalytic active sites for the activation of each of the intermediates with superior catalytic cyclic azo compounds.

Alternatively, loading 1.20 mmol $Na_2S$ to replace tripropylamine as electron donors gives the conversion of diamino compounds upon irradiation for 2 h, and no azo compounds were detected. Nevertheless, loading 0.30 mmol of $Na_2S$ under identical reaction conditions gave a 42% yield of the compound corresponding to 1-nitro-2-[2-(2-nitrophenoxy)ethoxy]benzene, and no monoamino or azo products were detected from the conversion. We inferred that the electron injection kinetics via the openings of capsule is essential for the stepwise reduction of nitro compounds into azo products. Only if the electron donation is slower than the condensation reaction of the two functional groups, azo products are formed, otherwise, amino compounds are selectively formed. The one-pot photocatalytic approach starts from easy-to-handle nitroarene feedstocks with diminished reaction steps and excellent selectivity, paving a new avenue for the rational design of photocatalytic manifolds.

We further expanded the manifolds for the preparation of asymmetric azo compounds, since most of the azo dyestuffs are asymmetric and the development of new, easy-to-handle strategies for preparation of asymmetric azo products from nitroarenes is still an urgent issue. The successful formation of asymmetric azo products could also provide evidences of the co-encapsulation of two different nitro substrates and the optimal electron injection kinetics for the direct reduction of nitroarenes into azo compounds in the microenvironment catalyst $H_F$. Nitrobenzene (0.025 mmol) was loaded in a reaction mixture containing $H_F$ (0.50 μmol), tripropylamine (0.60 mmol) and 0.025 mmol of 1-nitronaphthalene, 9-nitro-anthracene or 1-nitropyrene, respectively. As can be expected, asymmetric azo products, in addition to the two symmetric azo compounds were formed as major products under the identical conditions after irradiation for 6 h. The selectivity of the asymmetric azo product over the two symmetric azo products increased with increasing molecular size of the nitroarene substrates when nitrobenzene acted as one partner, which matched well with the size complementarity effect for the co-encapsulation for different substrates within $H_F$. The more than 90% selectivity of the asymmetric azo product demonstrates a new method that is different from the strategies that use bulk solutions and solid-state catalysts to prepare asymmetric azo products as fine chemicals (Fig. 8a).

Of the competing reactions with substrate N, N-dimethyl-4-nitroaniline (14b–16b), an important auxiliary to improve the properties of azo dyestuffs, asymmetric azo compounds were formed with moderate to excellent selectivity. Condensation reactions between N, N'-dimethyl-4-nitroaniline, and 2-nitrocarboxylbenzene or 2-nitro-4-sulfonicbenzene as the other partner yielded asymmetric azo products as the commercial methyl red and methyl orange, respectively, with selectivity up to 99%. In a reaction containing 1.0 mmol of N, N'-dimethyl-4-nitroaniline and 2-nitro-4- sulfonicbenzene, the loading of a 0.05% mole ratio of $H_F$ produced 285 mg

**Fig. 7 Photocatalyzed reduction of dinitro substrates using different reductants.** Isolated yields are reported. Standard conditions for cyclic azo product conversion were as follows: 1.0 mol% $H_F$ (0.50 µmol), substrate (0.05 mmol) and tripropylamine (1.20 mmol) under light irradiation for 10 h. Standard conditions for diamino product conversion were as follows: 1.0 mol % of $H_F$ (0.50 µmol), substrate (0.05 mmol) and $Na_2S$ (1.20 mmol) under light irradiation for 2 h.

(yield of 94%) of methyl orange, demonstrating the high stability of the catalytic systems.

Several catalysts have been reported so far for the selective reduction of nitroarenes into asymmetric azo products using thermal-driven and light-driven catalytic strategies[17,20]. $H_F$ represents the first example of microenvironment catalysts utilizing the inclusion thermodynamics of substrates to determine the selectivity of asymmetric azo products over the symmetric ones. In the previous reported hydrogenative coupling of nitroarenes, excess of the less reactive nitro substrate must be loaded into the cross-coupling reactions to kinetically balance off the self-condensation side reaction[17,18], favoured the trapping of the reactive-partner-generated azanol intermediate by the surface-abundant nitroso species derived from the less reactive partner to form the asymmetric azo products. In our new synthetic strategy, the slow entry kinetics of the electron donors with large estimated diameters were used to retard the electron injection kinetics and avoid the further reduction of the intermediates to the amino products, facilitating the selective condensation between the nitroso and azanol species.

We also recognized that the selectivity for the asymmetric azo products over symmetric azo compounds exhibited a positive correlation with the Gibbs free energy of the clathrates with the substrates, in the case of that the substrates have a similar estimated diameter with another partner (Supplementary Fig. 37b). Meanwhile, the selectivity for asymmetric azo products over symmetric azo compounds exhibited a positive correlation with the estimated diameter of the substrates in the capsule when the substrates did not comprise functional groups, except the nitro group, like that of the partner, nitrobenzene (Supplementary Fig. 37a). We envisioned that the asymmetric azo product $Ar_1−N=N−Ar_2$ was formed with a high probability from the condensation reaction between an $Ar_1−NO$ molecule with a smaller estimated diameter or higher entry kinetics to ensure the fast entrance, and an alternative $Ar−NHOH$ molecule with larger

retention time, or higher inclusion Gibbs free energy to ensure the higher stability of the corresponding clathrate. Thinking outside that the diffusion coefficient and inclusion constant should comprise of contributions of functional groups (i.e. $NO_2$, NO, NHOH) and aromatic fragments, we simply utilize the difference on inclusion free energy changes $−[\Delta G(Ar_2\text{-}NO_2)\text{-} \Delta G(Ar_1\text{-}NO_2)]$ between two nitroarenes to predict the selectivity of the asymmetric azo compound over the symmetric ones. In all cases, a rough linear relationship between the logarithm of $\text{yield}_{asym}/\text{yield}_{sym}$ and the difference in the free energy changes in the microenvironment of $H_F$ is proved (Fig. 8c), but the relationship between the selectivity and ratio of diffusion coefficient was not explicit in the case that the substrates having the similar molecular sizes (Fig. 8b).

Moreover, our enzymatic procedure exhibited laudable catalytic performance among the reported catalysts for the preparation of asymmetric azo products, including high catalytic activity, milder reaction conditions and size-dependent selectivity. As shown in Fig. 9, the reduction of nitroarenes provided excellent yields and selectivity over the byproducts, including amino products and symmetric azo compounds. From a possible mechanistic point of view, the slower electron injection rate of the enzymatic catalysis allowed the stepwise reduction of nitroarenes, wherein the in situ formed azanol species condensed with the nitroso partner to form azo compounds before the azanol species was further reduced to produce amino products or squeezed out of the capsule by other substrate molecules.

In summary, we reported a new enzymatic approach to the preparation of asymmetric and cyclic azo compounds from the one-pot reduction of nitro substrates under light irradiation. Simulating the reactant and substrate entry in natural enzymes, electron donors with large estimated diameters were utilized to retard the electron injection kinetics for the stepwise reduction of nitroarenes into nitroso and azanol species, followed by the condensation before the substrates were further reduced or

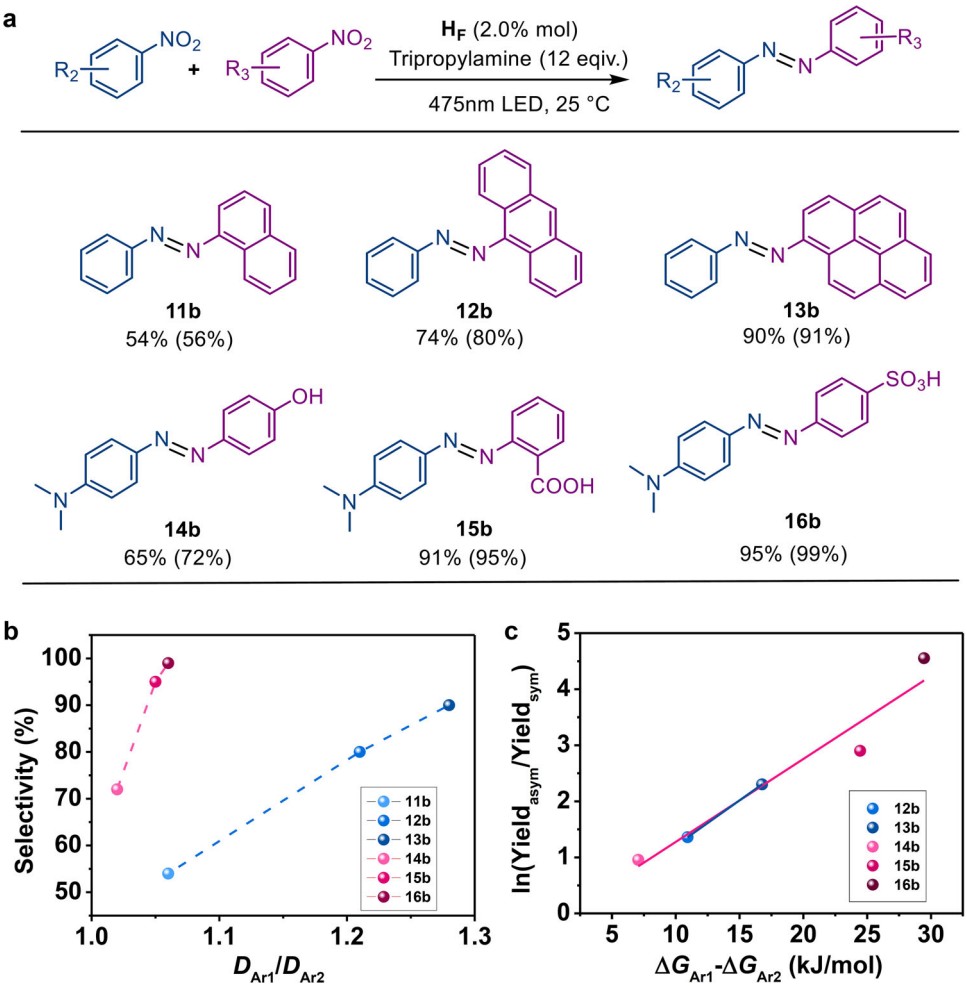

**Fig. 8 Selective condensation of nitroarenes into asymmetric azo products. a** Selective condensation of nitroarenes into asymmetric azo showing the isolated yields with the selectivity in parentheses under standard conditions: $H_F$ (0.5 µmol), equimolar amounts of nitroarenes substrates (0.025 mmol) and tripropylamine (0.60 mmol) for the irradiation of 6 h. **b** The relationship between the selectivity of the asymmetric azo products over the symmetric ones and the ratio of the diffusion coefficients corresponding to the nitroarenes. **c** The relationship between the yield ratio of the asymmetric azo compounds (yield$_{asym}$) over the symmetric ones (yield$_{sym}$) showing as ln(yield$_{asym}$/yield$_{sym}$) and the difference of Gibbs free energy changes corresponding to the two nitroarenes within the microenvironment catalyst $H_F$. (The Gibbs free energy changes corresponding to nitrobenzene and N, N'-dimethyl-4-nitroaniline were just used as references).

squeezed out of the microenvironment by other substrates. Our dye-loaded photocatalyst utilized the estimated diameter and inclusion Gibbs free energy first time to govern the selectivity for asymmetric azo products over asymmetric azo compounds, enabling the enzymatic one-pot procedure to exhibit laudable catalytic performance for the preparation of commercial asymmetric azo dyes. Competing inhibition experiments by adding ATP and the product into the reaction mixture support that the photocatalytic conversion occurred inside the cavity of $Co_6L_4^{6+}$ in a typical enzymatic fashion. The new approach that starts from easy-to-handle nitroarene feedstocks and endows excellent selectivity from one-pot photocatalytic conversion would pave a new avenue for rational design and optimization of photocatalysts toward green chemistry.

## Methods

**Materials and methods.** Unless otherwise specified, all chemicals were of reagent grade and were obtained from commercial sources and used without further purification. Elemental analyses: The elemental analyses of C, H and N were performed on a Vario EL III elemental analyzer. $^1$H NMR and $^{13}$C NMR spectra were recorded on a Varian INOVA-400M spectrometer. Isothermal titration calorimetry was performed on a Nano ITC. The solution fluorescence spectra were measured on an Edinburgh FS920 steady-state fluorescence spectrophotometer.

Cyclic voltammetry was performed on a CHI 660E electrochemical workstation with a three-electrode system using a Ag/AgCl electrode as the reference electrode, 0.5 mm diameter platinum silk as the counter electrode, and a glassy carbon electrode as the working electrode. The measurements were performed after degassing the solutions with argon to eliminate the effects of oxygen. The ESI mass spectra were all collected on an HPLC-Q-Tof-MS spectrometer in acetonitrile/methanol solution. Gas chromatography was characterized by GC 7890 T instrument analysis using a 5 Å molecular sieve column (0.6 m × 3.0 mm) with thermal conductivity detector, and argon as carrier gas. HPLC analysis was performed on a SHIMADZU LC 2030 Plus analyzer using a ZORBAX SB-C18 reversed-phase column (250 × 4.6 mm I. D, s-5 µM) and eluted with methanol and water to determine the yields of the catalytic reactions.

**Synthesis of H₃L.** Five drops of acetic acid were added to a mixture of 2-(diphenylphosphino)benzaldehyde (0.87 g, 3.0 mmol) and 4,4',4"-hydrazide- triphenylamine (0.42 g, 1.0 mmol) in a methanol solution. After the mixture was refluxed for 20 h, a white precipitate was formed and collected by filtration. Yield: 90%. Anal. Calc. for $C_{78}H_{60}N_7O_3P_3$: H, 4.89; C, 75.78; N, 7.93. Found: H, 4.96; C, 75.82; N, 7.82. $^1$H NMR (DMSO-d₆, 400 MHz, ppm): 11.98 (s, 3H), 9.17 (s, 3H), 8.08 (s, 3H), 7.90 (d, 6H), 7.52–7.48 (t, 3H), 7.42–7.37 (m, 21H), 7.24–7.15 (m, 18H), 6.86–6.83 (m, 3H). $^{13}$C NMR (100 MHz, DMSO-d₆) δ 162.86, 149.53, 145.62, 138.37, 136.61, 135.99, 133.99, 133.80, 133.70, 130.57, 129.66, 129.44, 129.37, 128.70, 126.22, 123.99.

**Synthesis of H₁.** H₃L (123.5 mg, 0.10 mmol) and CoCl₂·6H₂O (35.7 mg, 0.15 mmol) were mixed in a methanol solution and refluxed for 12 h. Then,

**Fig. 9 Scope of selective conversion of asymmetric azo compounds.** Isolated yields are reported with the selectivity in parentheses. Standard conditions for asymmetric azo product conversion: 2.0 mol% $H_F$, two substrates with equimolar amounts (0.025 mmol) and tripropylamine (0.60 mmol) under light irradiation for 6 h.

$NH_4PF_6$ (24.5 mg, 0.15 mmol) was added, and the solution was stirred for another 1 h. After the reaction mixture was cooled to room temperature, a dark red precipitate formed and was collected by filtration. Yield: 70%. Dark red crystals of $H_1$ were obtained by slowly diffusing diethyl ether into a DMF solution of $H_1$. Anal. Calc. for $C_{375}H_{401}Co_6F_{24}N_{38}O_{40}P_{16}$: H, 5.46; C, 60.89; N, 7.28. Found: H, 5.50; C, 61.14; N, 7.31. $^1H$ NMR ($CD_3CN$, 400 MHz, ppm): 9.15 (s, 3H), 8.00–7.98 (m, 3H), 7.88–7.84 (m, 3H), 7.55–7.40 (m, 6H), 7.25–6.98 (m, 39H), 6.82–6.80 (m, 3H). $^{13}C$ NMR (100 MHz, $CD_3CN$) δ 167.06, 154.71, 145.03, 137.92, 137.40, 136.44, 133.75, 130.85, 128.73, 128.68, 128.51, 128.23, 121.60. ESI–MS: m/z: 881.02, $[(Co_6L_4)]^{6+}$ and 1086.24, $[(Co_6L_4)·PF_6]^{5+}$.

Crystal data for $H_1$. $C_{375}H_{401}Co_6F_{24}N_{38}O_{40}P_{16}$, Mr = 7406.93, monoclinic space group C2/c, dark-red block, $a = 27.2433$ (8) Å, $b = 48.4238$ (14) Å, $c = 35.3446$ (10) Å, $β = 111.329(2)°$, $V = 43424(2)$ Å$^3$, $Z = 4$, Dc = 1.133 gcm$^{-3}$, $T = 173(2)$ K, μ(Mo-Kα) = 0.355 mm$^{-1}$. For 38126 unique reflections ($R_{int} = 0.1087$), final $R_1$ [with $I > 2σ(I)$] = 0.0903, $wR_2$ (all data) = 0.1674, GOOF = 0.960. CCDC NO. 1913942. For the refinement of $H_1$, except for those of the solvent molecules, the non-H atoms were refined anisotropically, and the hydrogen atoms were fixed geometrically at calculated distances and allowed to ride on the parent nonhydrogen atoms. Two of the benzene rings in the ligands and several F atoms in the counter $PF_6^-$ ions were disordered into two parts with the site occupied factors of each part being fixed as 0.5, respectively. The related bond distances in the solvent molecules were restrained as idealized values. The thermal parameters of the adjacent atoms of several solvent molecules were restrained to be similar.

In the checkcif file, the A alert is caused by the weak diffraction intensity of the poor quality crystal, and the "short inter D…A contact" is caused by the presence of partly occupied solvent molecules with highly disordered atoms. The crystal analysis here could offer the structure of the cage compound and estimate the size of the cavity, which is helpful for the research of the catalytical chemistry of the cage compound.

**General procedure for catalytic reactions.** All catalytic reactions were carried out in a homemade 25.0 mL flask containing 5.0 mL acetonitrile/water 4/1 (v/v) solution with the pH of the reaction solution adjusted to 10.0. The solution was degassed by blowing Ar gas for 20 min and irradiated under 475 nm LED (18 mW/cm$^2$) to ensure excitation of only the photosensitizer. In addition, the reaction temperature was maintained at 25 °C by circulating water through the outer packet of the reactor. The yields of the isolated azo products were determined by $^1H$ NMR, and the reactions were traced through HPLC on a SHIMADZU LC 2030 Plus analyzer.

The photocatalytic reactions of individual nitro substrates reduction using $Na_2S$ as electron donor were carried out in a 25 mL homemade flask containing $H_F$ (1.0 mol%, 0.50 μmol), substrate (nitrobenzene, 1-nitronaphthalene, 9-nitroanthracene, or 1-nitropyrene) (0.05 mmol), $Na_2S$ (0.60 mmol) in 5.0 mL acetonitrile/water 4/1 (v/v) solution. The pH of the reaction solution was adjusted to 10.0 and degassed by blowing Ar gas for 20 min and irradiated under 475 nm LED. The analysis of the yield and the formation rate of the product was performed on a SHIMADZU LC 2030 Plus analyzer. The tracking of the reaction process was

carried out by extraction of 50 μL reaction mixture at a certain time with a long needle and followed by HPLC analysis after the quick flush through the silica gel. For substrate 2,4,6-triphenylnitrobenzene, the generated photoproduct $H_2$ was characterized on a GC 7890 T instrument. The amount of hydrogen was determined using an external standard.

Kinetics experiments were carried out in a 25 mL homemade flask containing $H_F$ (0.50 μmol), $Na_2S$ (1.20 mmol), and nitro substrate (0.1 mmol, 0.05 mmol, 0.025 mmol, and 0.005 mmol, respectively) in 5.0 mL acetonitrile/water 4/1 (v/v) solution. The pH of the reaction solution was adjusted to 10.0 and was then degassed by blowing Ar gas for 20 min and irradiated under 475 nm LED. The yields of amino product in the first 5 min were regarded as the initial rate of the reactions, which were further used in the linear fitting.

The photocatalytic reactions of competing reduction of nitro substrates were also carried out using $Na_2S$ as reductant. In a 25 mL flask, $H_F$ (0.50 μmol), substrate nitrobenzene (0.05 mmol), another nitro substrate (1-nitronaphthalene, 9-nitroanthracene, or 1-nitropyrene) (0.05 mmol), and $Na_2S$ (0.60 mmol) were added into 5.0 mL acetonitrile/water 4/1 (v/v) solution (pH = 10.0), and then the mixture was degassed by blowing Ar gas for 20 min. After that, the mixture was irradiated under 475 nm LEDs. The formation of amino products was analysis using SHIMADZU LC 2030 Plus analyzer. The tracking of the reaction process was carried out by extraction of 50 μL reaction mixture at a certain time with a long needle and followed by HPLC analysis after the quick flush through the silica gel.

The photocatalytic reactions of 9-nitroanthracene reduction using different reductants were carried out in a 25 mL homemade flask containing $H_F$ (1.0 mol%, 0.50 μmol), 9-nitroanthracene (0.05 mmol), and reductant (N,N-dimethyl-benzenamine, triethylamine, N-phenylpyrolidine, N,N-diethyl- benzenamine, and tripropylamine, respectively) (0.60 mmol) in 5 mL acetonitrile /water 4/1 (v/v) solution. The pH of the reaction solution was adjusted to 10.0 and was then degassed by blowing Ar gas for 20 min. After 6 h irradiation under 475 nm LED, the yields of amine and azo products were analyzed on a SHIMADZU LC 2030 Plus analyzer.

The photocatalytic 9-nitroanthracene reduction into 9-nitrosoanthracene using different reductants were also carried out in the same conditions, only the addition of reductants was fixed at 0.10 mmol.

The photocatalytic reactions of selectivity formation of asymmetric azo compounds from two different nitro substrates were carried out using tripropylamine as reductant. In a 25 mL homemade flask, $H_F$ (0.50 μmol), tripropylamine (0.60 mmol), substrate $Ar_1$ (nitrobenzene, or N,N-dimethyl-4-nitrobenzene) (0.025 mmol), and substrate $Ar_2$ (1-nitronaphthalene, 9-nitroanthracene, 1-nitropyrene, 4-methyl- nitrobenzene, 4-hydroxynitrobenzene, 2-carboxynitrobenzene, or 4-sulfonic- nitrobenzene) (0.025 mmol) were added in 5.0 mL acetonitrile/water 4/1 (v/v) solution. The pH of the reaction solution was also adjusted to 10.0 and was then degassed by blowing Ar gas for 20 min. After 6 h irradiation under 475 nm LED, the reaction solution was analyzed on a SHIMADZU LC 2030 Plus analyzer after the quick flush through the silica gel.

For substrates containing two nitro groups and a flexible chain, the photocatalytic reactions for the formation of diamino product were carried out

using $Na_2S$ as reductant. In a 25 mL homemade flask, $H_F$ (1.0 mol%, 0.50 μmol), $Na_2S$ (1.20 mmol), substrate (0.05 mmol), were added in 5 mL acetonitrile/water 4/1 (v/v) solution. The pH of the rection solution was also adjusted to 10.0 and was then degassed by blowing Ar gas for 20 min. After 2 h irradiation under 475 nm LED, the yields of the reactions were analyzed by $^1H$ NMR.

For substrates containing two nitro groups and a flexible chain, the photocatalytic reactions for the formation of cyclic diazocines products were carried out using tripropylamine as reductant. In a 25 mL homemade flask, $H_F$ (1.0 mol%, 0.50 μmol), tripropylamine (1.20 mmol), substrate (0.05 mmol), were added in 5 mL acetonitrile/water 4/1 (v/v) solution. The pH of the rection solution was also adjusted to 10.0 and was then degassed by blowing Ar gas for 20 min. After 10 h irradiation under 475 nm LED, the yields of the reactions were analyzed by $^1H$ NMR.

## Data availability

The X-ray crystallographic coordinates for the structure reported in this article have been deposited at the Cambridge Crystallographic Data Centre (CCDC) under the deposition number CCDC 1913942. The data can be obtained free of charge from the Cambridge Crystallographic Data Centre via http://www.ccdc.cam.ac.uk/data_request/cif. The structure of fluorescein (Fl) is downloaded from PDB database via https://www.rcsb.org/. All other data supporting the findings of this study are available within the article and its Supplementary Information files or from the corresponding author upon request.

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

## Acknowledgements

This work was supported by the National Natural Science Foundation of China (21890381, 21971030 and 21820102001).

## Author contributions

Y.Y., X.J., and C.Y.D. conceived the project and designed the experiments. Y.Y. carried out the main experiments, collected and interpreted the data. Y.Y., J.Z, F.Y prepared the ligand. Y.Y., X.J., and C.Y.D. cowrote the paper. All authors discussed the results and commented on the manuscript.

## Competing interests

The authors declare no competing interests.
