## [Peer Review File · Nature Communications]

REVIEWER COMMENTS

Reviewer #1 (Remarks to the Author):

The manuscript titled "Modifying electron injection kinetics of a microenvironment catalyst for selective photoreduction of nitroarenes into cyclic and asymmetric azo compounds" by Yang and coworkers reports an enzyme-like approach that regulates the reactants and substrates entry behavior to control the electron injection kinetics and the products selectivity of the nitroarenes photoreduction. Moreover, the estimated diameter and inclusion Gibbs free energy of the substrates were utilized first time to control and predict the products selectivity. This work is well-organized and all experiments are well described, and it offers new perspectives to rational design and optimization of photocatalysts in biomimetic system. This reviewer believes this work will attract attentions from a broad community of readers of Nature Comm and recommends its publication after the following minor issues being addressed:

1. The 1:2 binding between Hf and nitroarene substrates is an important factor for highly selective formation of the reductive-coupling Aza products. While it is good to see that ITC experiment indicated a 1:2 binding ratio, this reviewer think it will definitely enhance such a conclusion if other spectroscopic data, such as NMR or MS are discussed concerning these host-guest complexes.
2. One big challenge in supramolecular catalysis is the product inhibition. In this work, it seems the larger Azo compounds were in-situ formed and then squeezed out of the capsule rather than blocking the host cavity. It will be helpful if the binding affinities are compared between the substrates and the products with the host.
3. The authors observed that "The selectivity of the asymmetric azo product over the two symmetric azo products increased with increasing molecular size of the nitroarene substrates when nitrobenzene acted as one partner". Here, the size complementarity effect for the co-encapsulation for different substrates within Hf may be another explanation to be considered.

Reviewer #2 (Remarks to the Author):

The authors developed a supramolecular catalytic system with a coordination cage, a dye molecule, a reductant, and nitro substrates. The kinetics of photo-induced reduction is controlled to realize selective azo product formations. The attainment of highly selective reactions by fine-tuning of inclusion kinetics and thermodynamics employs distinctive characters of a cage-like host molecule. Therefore, it can be considered as an ideal form of supramolecular host catalysts. The system showed good catalytic activity and substrate scope, including cyclic, acyclic, symmetric, and asymmetric azo products, showing the usefulness of the system as a practical synthetic method. The authors report careful experiments to show how supramolecular catalysis works. Overall, this work is recommended for publication after some revisions. I recommend the authors consider the following points.

1. For readers who are not familiar with the photo-induced reaction, the reaction cycle is hard to be understood. In the first step, which part of the system absorbs light and is excited to give an electron (or energy) to which part? How does the electron on the reductant reaches nitro substrates? Please provide a schematic explanation of the flow of energy or electrons in Figure 1.
2. The position of fluorescein in the system is still controvertible. The authors claim the encapsulation of fluorescein within the cavity of the cage, but the chemical shift changes of the protons on fluorescein and the host molecule are subtle. In general, encapsulation of guest molecules in a cavity surrounded by aromatic components causes significant upfield shifts of the guest signals in ^1H NMR. Another interpretation of the result would be interactions between the fluorescein molecule and the outer part of the host.

3. How do the interactions between electron donors and the host cavity work? The authors discussed the effective diameter of the donors, but it is unclear whether the kinetics of entering the host cavity through portals is essential. Averaged distances between the donors and the host or substrates determined by steric hindrance are possible to affect. Is there any evidence for the binding?

4. The authors used Hill plots to reveal the host-guest complexation ratios. Although the plots can determine the cooperative behavior in the binding, they cannot determine the host-guest ratio (J. N. Weiss, FASEB J. 1997, 11, 835.). In principle, Job plot is required to determine the ratio.

5. The authors explain the origin of selectivity in two ways: the estimated diameters and the inclusion free energy changes. However, two sets of parameters were separately provided for different groups of substrates. For example, the authors used the former to explain selectivities in the reaction between apolar substrates, but the latter for polar substrates. The readers might feel difficulty in grabbing the key point for substrate choice. Can the authors explain them by the same parameter?

6. I agree that one substrate should be tightly bound in the cavity, and the other should quickly enter the cavity to control the reaction kinetics. However, it is expected that the stronger binding of one substrate also facilitates homo-dimerization through their bimolecular encapsulation. Why can such a side reaction be suppressed?

Reviewer #3 (Remarks to the Author):

The authors reported the synthesis of the octahedral cage Co6L46+, which can catalyze the one-pot reduction of nitro substrates to asymmetric and cyclic azo compounds under light irradiation when loading dye in the cage. The topic is very interesting, and the experiments are sufficient and extensively discussed. However, some issues need to be further addressed and clarified to improve the chemistry.

1. In Size-dependent Photoreduction of Nitroarenes section, more control experiments other than the conditions in absence of any component (HF, substrate and tripropylamine) are needed, for example, replacing HF with just H or F, replacing H with its organic fragment or simple metal-complexes, which may provide information if the cage and hosted dye are necessary for the reaction instead of the simply mixing of relative chemical components.

2. In Supplementary Figure 13, shifts of the signals corresponding to OH protons are easily affected by solvent conditions and the signals of the cage show no shifts, which cannot indicate the encapsulation of fluorescein. An NMR titration spectrum is needed to confirm the conclusion.

3. Formation of pure and solution stable guest encapsulated HF as the catalyst is one of most important bases in this research. Supplementary experiments like DOSY, ¹H NMR titration are necessary to check the host-guest chemistry. Is there slow or fast exchange of the guests?

4. A major concern is about the complex host-guest chemistry under conditions of catalysis. Individually, author studied the complexation of host plus fluorescein dye (1:1) and host plus nitrobenzenes (1:2 or 1:1 in case of 1-nitropyrene), however, how about the host-guest chemistry in the presence of both dye substrate molecules? Is there any influence between each other?

5. Along this line, how about the influence from electron donor agents like N,N'-dimethylaniline, triethylamine, 1-phenylpyrrolidine, N,N'-diethylaniline or tripropylamine? The crystal structure shows many disordered small molecules inside the cage, so is there any evidence that these molecules cannot enter the cage cavity? This is an important issue because they are considered to determine "electron injection kinetics".

6. Similar concern is about how to practically control "modifying electron injection kinetics of a microenvironment catalyst". The diffusion coefficients of the mentioned electron donors are only

slightly different (SI Table 3). It is also confusing in discussion like "As a larger molecular size of electron donors always results in slower entry kinetics or electron injection kinetics into the microenvironment of HF through the openings, a smaller estimated diameter (D) with faster electron injection kinetics of the electron donor should allow the rapid and continuous reduction of the substrate and intermediates". How the size of these donor molecules impact the entry kinetics or electron injection kinetics, through ingress of donor molecules into cage or other reason? If the kinetics is related to the exchange rate of the substrates, reduction actually happens outside the cage in solution.

Other minor issues:

- 1. Some typos need to be corrected, such as "Slective" should be changed to "Selective".**
- 2. Supporting information, page 71, there is an NMR spectrum showing different format from others.**
- 3. The checkcif shows so many lever A/B alerts. The authors should add hydrogen for the disorder solvent molecules and solve or explain other level A/B alerts.**

Reviewer: 1**Reviewer's Comments:**

The manuscript titled "Modifying electron injection kinetics of a microenvironment catalyst for selective photoreduction of nitroarenes into cyclic and asymmetric azo compounds" by Yang and coworkers reports an enzyme-like approach that regulates the reactants and substrates entry behavior to control the electron injection kinetics and the products selectivity of the nitroarenes photoreduction. Moreover, the estimated diameter and inclusion Gibbs free energy of the substrates were utilized first time to control and predict the products selectivity. This work is well-organized and all experiments are well described, and it offers new perspectives to rational design and optimization of photocatalysts in biomimetic system. This reviewer believes this work will attract attentions from a broad community of readers of Nature Comm and recommends its publication after the following minor issues being addressed.

Q1: The 1:2 binding between \mathbf{H}_F and nitroarene substrates is an important factor for highly selective formation of the reductive-coupling Aza products. While it is good to see that ITC experiment indicated a 1:2 binding ratio, this reviewer think it will definitely enhance such a conclusion if other spectroscopic data, such as NMR or MS are discussed concerning these host-guest complexes.

Responses: Thank you for your good suggestion. We have added the MS data of the \mathbf{H}_F upon addition of substrate 4-nitrophenol (**S**), which exhibited the species of $[\mathbf{H}_F \cdot 2\mathbf{S} \cdot \text{PF}_6]^{5+}$ at $m/z = 1208.2942$, showing a 1:2 binding ratio between \mathbf{H}_F and **S** that coincides with the results of ITC experiment. We have added the relevant experimental data and some words concerning the clathrates in the revised supporting information (Supplementary Fig. 5) and the main text, respectively.

Q2: One big challenge in supramolecular catalysis is the product inhibition. In this work, it seems the larger Azo compounds were in-situ formed and then squeezed out of the capsule rather than blocking the host cavity. It will be helpful if the binding affinities are compared between the substrates and the products with the host.

Responses: Thank you for your suggestion. We added few of sentences corresponding to the product inhibition in the main text of the manuscript using 4-nitrophenol as the model substrate, as the hydroxyl group worked as an anchoring site to ensure the host-guest interactions being strong enough for easily detection. Adding up to 0.025 mmol of the azo product 4,4'-dihydroxyazobenzene into the reaction mixture containing \mathbf{H}_F (0.50 μmol), 4-nitrophenol (0.05 mmol) and tripropylamine (0.60 mmol) only caused slight decrease in the conversion of azo compounds under the standard conditions. Also, the initial rate of the photoreduction of 4-nitrophenol remained unchanged with the increasing concentrations of 4,4'-dihydroxyazobenzene. Upon addition of the substrate, reduction intermediates and products into the \mathbf{H}_F solution, isothermal titration calorimetry experiments demonstrated that the formation of clathrate with the substrate 4-nitrophenol exhibited higher Gibbs free energy

change (-25.26 kJ/mol) than that of the azo product 4,4'-dihydroxyazobenzene (-24.27 kJ/mol), but with different inclusion number. These results indicated that the substrate could probably squeeze out the product and keep the reaction proceeding continuously.

Q3: The authors observed that "The selectivity of the asymmetric azo product over the two symmetric azo products increased with increasing molecular size of the nitroarene substrates when nitrobenzene acted as one partner". Here, the size complementarity effect for the co-encapsulation for different substrates within H_F may be another explanation to be considered.

Responses: Thank you for your suggestion. We agree well with that the size complementarity effect for the co-encapsulation for different substrates within H_F was important to control the formation of asymmetric azo products from two different nitroarenes. In fact, in our original manuscript, we mentioned that in the presence of two different nitroarenes, the H_F endows favorable condensation between an azanol partner with larger retention kinetics in the capsule and a nitroso partner with larger entrance kinetics into the capsule. This is another explanation of the size complementarity effect. We added some words about this special and interesting performance of the microenvironment catalysts in the revised manuscript.

Reviewer: 2**Reviewer's Comments:**

The authors developed a supramolecular catalytic system with a coordination cage, a dye molecule, a reductant, and nitro substrates. The kinetics of photo-induced reduction is controlled to realize selective azo product formations. The attainment of highly selective reactions by fine-tuning of inclusion kinetics and thermodynamics employs distinctive characters of a cage-like host molecule. Therefore, it can be considered as an ideal form of supramolecular host catalysts. The system showed good catalytic activity and substrate scope, including cyclic, acyclic, symmetric, and asymmetric azo products, showing the usefulness of the system as a practical synthetic method. The authors report careful experiments to show how supramolecular catalysis works. Overall, this work is recommended for publication after some revisions. I recommend the authors consider the following points.

Q1: For readers who are not familiar with the photo-induced reaction, the reaction cycle is hard to be understood. In the first step, which part of the system absorbs light and is excited to give an electron (or energy) to which part? How does the electron on the reductant reaches nitro substrates? Please provide a schematic explanation of the flow of energy or electrons in Figure 1.

Responses: Thank you for your good suggestion. We have added a schematic explanation of the electron transformation in Fig. 1b in the revised manuscript. During the reaction of photocatalytic cycle, the **FI** molecule that encapsulated into the cavity absorbs light and triggers the photoinduced electron transfer (PET) from the excited state of **FI** to Co ions in **H₁**. The reduced Co centers donated the proton and electron to the nitro substrates encapsulated *via* a proton coupled electron transfer process (PCET). The electron donor injection electron to the *in situ* formed oxidized photosensitizer to recovery the organic dye and the micro-environment catalyst **H_F** for another cycle of electron donation and final generated the product. In case of the electron donor was quite large, the electron injection kinetics is retarded by the diffusion kinetics of electron donor into the microenvironment, the *in situ* formed nitroso and azanol intermediates condensed each other first, before they were further reduced or squeezed out of the microenvironment by other substrate or intermediates.

Q2: The position of fluorescein in the system is still controvertible. The authors claim the encapsulation of fluorescein within the cavity of the cage, but the chemical shift changes of the protons on fluorescein and the host molecule are subtle. In general, encapsulation of guest molecules in a cavity surrounded by aromatic components causes significant upfield shifts of the guest signals in ¹H NMR. Another interpretation of the result would be interactions between the fluorescein molecule and the outer part of the host.

Responses: Thank you for your good suggestion. We dried the deuterated solvent and preprocessed the **H₁** sample with supercritical carbon dioxide extraction to avoid the influence of solvent molecules that related to the structure of **H₁**. Then an ¹H NMR titration experiment between **H₁** and **FI** was carried out by adding **FI** successively into a 1.0 mM **H₁** solution (Supplementary Fig. 16). Small but obvious downfield shift corresponding to OH protons in

FI was observed, along with the upfield shifts corresponding to aromatic protons in **FI**, indicating the encapsulation between **H₁** and **FI**. For better understanding of the formation of **H_F** and the position of **FI**, we have taken the docking calculations to confirm the theoretically optimization of the clathrate structure of **H_F**. As shown in Fig. 2b, the encapsulated **FI** molecule is stuck at one of the openings of **H₁** and it occupies only a little interior cavity, beneficial the encapsulation of other guest molecules, including the substrates and electron donors. We added some words about the ¹H NMR titration even for different temperature.

Q3: How do the interactions between electron donors and the host cavity work? The authors discussed the effective diameter of the donors, but it is unclear whether the kinetics of entering the host cavity through portals is essential. Averaged distances between the donors and the host or substrates determined by steric hindrance are possible to affect. Is there any evidence for the binding?

Responses: Thank you for your good question. We investigate the possible interactions between the electron donors and microenvironment catalyst **H_F** using luminescence titration. Upon the addition of the same proportion of the electron donors according to the photocatalytic reaction to the same **H_F** solution, our experimental results (Supplementary Fig. 29a) revealed that N, N'-dimethylaniline was the strongest quencher, while tripropylamine was the weakest one. We attributed the emission quenching to the interactions between the microenvironment catalyst **H_F** and the electron donors. The positive relationship of the yield ratio between the azo and the emission quenching efficiency of electron donors for the reduction conversion further support our postulation and verify interactions between the electron donors and the microenvironment catalyst.

Meanwhile, as **H_F** only encapsulated one 1-nitropyrene molecule and 1-aminopyrene was the only photoreduction product. The initial conversion rate 1-aminopyrene with different electron donors were used to reveal the electron injection rates of the corresponding electron donor. As shown in Supplementary Fig. 29b, the positive correlation between the initial rate of the conversion and the estimated diameter of electron donors demonstrated that the entrance kinetics of electron donors into **H_F** dominated the electron injection kinetics and the photoreduction conversion rate.

Q4: The authors used Hill plots to reveal the host-guest complexation ratios. Although the plots can determine the cooperative behavior in the binding, they cannot determine the host-guest ratio (J. N. Weiss, FASEB J. 1997, 11, 835.). In principle, Job plot is required to determine the ratio.

Responses: Thank you for your good suggestion. We then used Job plot fitting to determine the ratios of host-guest complexation ratios between **H₁** and nitro substrates, or between **H_F** and nitro substrates (Supplementary Fig. 13). The results were similar to the previous fitting results and also fitted well with the microcalorimetric titration experiments.

Q5: The authors explain the origin of selectivity in two ways: the estimated diameters and the inclusion free energy changes. However, two sets of parameters were separately provided for

different groups of substrates. For example, the authors used the former to explain selectivities in the reaction between apolar substrates, but the latter for polar substrates. The readers might feel difficulty in grabbing the key point for substrate choice. Can the authors explain them by the same parameter?

Responses: Thank you for your good question. In the presence of different nitroarenes, the condensation reaction of which nitrosoarene is controlled by the associate kinetics (k_a) of nitroso species that entered the microenvironment, and of which azanolarene is dominated by the retention or dissociate kinetics (k_d) of the azano species. An asymmetric azo compound $Ar_1-N=N-Ar_2$ was formed with a high probability from a condensation reaction of Ar_1-NO having higher diffusion kinetics with Ar_2-NHOH having larger retention times and inclusion constant (K_{ass}). In a very preliminary model, the selectivity of an asymmetric azo compound over the corresponding symmetric ones was assumed to be governed by the ratio between associate rate constants of the two nitroso intermediates $k_a(Ar_1-NO) / k_a(Ar_2-NO)$ and the ratio between inclusion constant of the two azanol intermediates $K_{ass}(Ar_2-NHOH) / K_{ass}(Ar_1-NHOH)$. As the associate rate constant of the substrate-loaded clathrate is mainly controlled by the diffusion coefficient (D) of substrate itself, the selectivity should be controlled by the ratio of diffusion coefficients $D_{(Ar_1-NO)} / D_{(Ar_2-NO)}$ and the difference on inclusion free energy changes $-\Delta G_{(Ar_2-NHOH)} - \Delta G_{(Ar_1-NHOH)}$, according to the equation of that $\Delta G = -RT \ln K_{ass}$.

Thinking outside that the diffusion coefficient and inclusion constant should comprise of contributions of functional groups (*i.e.* NO_2 , NO , $NHOH$) and aromatic fragments, we simply utilize the difference on inclusion free energy changes, $-\Delta G_{(Ar_2-NO_2)} - \Delta G_{(Ar_1-NO_2)}$ between two nitroarenes to predict the selectivity of the asymmetric azo compound over the symmetric ones. In all cases, a rough linear relationship between the logarithm of $yield_{asym}/yield_{sym}$ and the difference in the free energy changes in the microenvironment of H_F is proved (Fig. 7b). We added some words about this special and interesting performance of the microenvironment catalysts in the revised manuscript.

Q6: I agree that one substrate should be tightly bound in the cavity, and the other should quickly enter the cavity to control the reaction kinetics. However, it is expected that the stronger binding of one substrate also facilitates homo-dimerization through their bimolecular encapsulation. Why can such a side reaction be suppressed?

Responses: Thank you for your good question. As mentioned above, in the presence of two different nitroarenes, the asymmetric azo product $Ar_1-N=N-Ar_2$ was formed with a high probability from a condensation reaction of Ar_2-NHOH having larger retention times and inclusion constant (K_{ass}) with Ar_1-NO having higher diffusion kinetics. Accordingly, we proposed that the symmetric azo compounds could be suppressed by enlarged the difference on the Gibbs free energy changes of the two substrates in the same microenvironment catalyst H_F . We simply utilize the difference on inclusion Gibbs free energy changes, $-\Delta G_{(Ar_2-NO_2)} - \Delta G_{(Ar_1-NO_2)}$ between two nitroarenes to predict and control the selectivity of the asymmetric azo compound over the symmetric ones.

Meanwhile, as the formation of azo compounds required the encapsulation of two substrate molecules into one microenvironment of H_F , the substrate partners with strong

binding ability to H_F , like 2-nitrocarboxylbenzene or 2-nitro-4-sulfonicbenzene, only formed 1:1 host-guest systems with H_F , the side homo-dimerization corresponding to these substrates was suppressed automatically. In the presence of other nitroarenes that are able to enter the microenvironment of H_F with high entry rate, asymmetric azo compounds would be achieved selectively.

Reviewer: 3

Reviewer's Comments:

The authors reported the synthesis of the octahedral cage $\text{Co}_6\text{L}_4^{6+}$, which can catalyze the one-pot reduction of nitro substrates to asymmetric and cyclic azo compounds under light irradiation when loading dye in the cage. The topic is very interesting, and the experiments are sufficient and extensively discussed. However, some issues need to be further addressed and clarified to improve the chemistry.

Q1: In size-dependent photoreduction of nitroarenes section, more control experiments other than the conditions in absence of any component (\mathbf{H}_F , substrate and tripropylamine) are needed, for example, replacing \mathbf{H}_F with just \mathbf{H} or \mathbf{Fl} , replacing \mathbf{H} with its organic fragment or simple metal-complexes, which may provide information if the cage and hosted dye are necessary for the reaction instead of the simply mixing of relative chemical components.

Responses: Thank you for your good suggestion. The control experiments were redesigned and carried out with just \mathbf{H}_1 , \mathbf{Fl} , the mononuclear complex \mathbf{M}_1 , \mathbf{M}_1 with addition of \mathbf{Fl} , the mixture of the ligand and CoCl_2 , the mixture of the ligand and CoCl_2 with addition of \mathbf{Fl} , CoCl_2 or CoCl_2 with addition of \mathbf{Fl} to replace \mathbf{H}_F under the conditions using Na_2S or tripropylamine as electron donors respectively. Also, the influence of the light was also considered in the control experiments. The mononuclear \mathbf{M}_1 (Supplementary Fig. 6) was synthesized by refluxing cobalt salts and (2-(diphenylphosphanyl) benzylidene) benzene carbohydzide, which resembled a corner of \mathbf{H}_1 and exhibited the identical cobalt-based redox property. To guarantee the equal amounts of cobalt ions, the amount of the addition of \mathbf{M}_1 or cobalt salts in the control experiments was six times as much.

As shown in Supplementary Table 4, under the standard conditions, the using of only \mathbf{H}_1 , \mathbf{Fl} , \mathbf{M}_1 , H_3L with CoCl_2 and CoCl_2 with the same proportion to that of the photoreaction mixture, only gave minimal aniline product. The using \mathbf{M}_1 (3.0 μmol) and the mixture containing H_3L and CoCl_2 resulted in 32% and 11% yields of aniline, respectively. Only minimal aniline product could be observed in the dark under varies conditions. These results indicated that the superiority of the dye-loaded microenvironment catalyst \mathbf{H}_F for the reduction of nitrobenzene into the amino product.

In the case of tripropylamine as electron donor, the utilization of only \mathbf{H}_1 , \mathbf{Fl} , \mathbf{M}_1 , H_3L with CoCl_2 and CoCl_2 with the same proportion to that of the photoreaction mixture to replace the microenvironment \mathbf{H}_F , did not lead to the conversion of substrate. The using of \mathbf{M}_1 and the mixture containing H_3L and CoCl_2 resulted little of aniline, but could not give the azo product. No reduction azo product could be observed in the dark under varies conditions. These results indicated the encapsulation of more than one substrate molecules in the microenvironment of \mathbf{H}_F was essential for the condensation of substrates into azo compounds.

Q2: In Supplementary Figure 13, shifts of the signals corresponding to OH protons are easily affected by solvent conditions and the signals of the cage show no shifts, which cannot indicate the encapsulation of fluorescein. An NMR titration spectrum is needed to confirm the conclusion.

Responses: Thank you for your good suggestion. We dried the deuterated solvent and preprocessed the \mathbf{H}_1 sample with supercritical carbon dioxide extraction to avoid the influence of solvent molecules that related to the structure of \mathbf{H}_1 . Then an NMR titration experiment between \mathbf{H}_1 and \mathbf{FI} was carried out by adding \mathbf{FI} successively into a 1.0 mM \mathbf{H}_1 solution. As shown in Supplementary Fig. 16, small but obvious downfield shift corresponding to OH protons in \mathbf{FI} was observed, along with the upfield shifts corresponding to aromatic protons in \mathbf{FI} , indicating the encapsulation between \mathbf{H}_1 and \mathbf{FI} .

Q3: Formation of pure and solution stable guest encapsulated \mathbf{H}_F as the catalyst is one of most important bases in this research. Supplementary experiments like DOSY, ^1H NMR titration are necessary to check the host-guest chemistry. Is there slow or fast exchange of the guests?

Responses: Thank you for your good suggestion. We dried the deuterated solvent and preprocessed the \mathbf{H}_1 sample with supercritical carbon dioxide extraction to avoid the influence of solvent molecules that related to the structure of \mathbf{H}_1 . Then an NMR titration experiment between \mathbf{H}_1 and \mathbf{FI} was carried out by adding \mathbf{FI} successively into a 1.0 mM \mathbf{H}_1 solution. Small but obvious downfield shift corresponding to OH protons in \mathbf{FI} was observed, along with the upfield shifts corresponding to aromatic protons in \mathbf{FI} , indicating the encapsulation between \mathbf{H}_1 and \mathbf{FI} . We also measured the low temperature NMR spectra. Comparing with the ^1H NMR spectra of the mixture of \mathbf{H}_1 and \mathbf{FI} at 298 K, the spectra at 273 K showed that the peaks of \mathbf{FI} and \mathbf{H}_1 were all broadened and the signal of H_a downfield shifted more significant, which indicated the existing of the slow exchange of guests (Supplementary Fig. 17).

Q4: A major concern is about the complex host-guest chemistry under conditions of catalysis. Individually, author studied the complexation of host plus fluorescein dyne (1:1) and host plus nitrobenzenes (1:2 or 1:1 in case of 1-nitropyrene), however, how about the host-guest chemistry in the presence of both dyne substrate molecules? Is there any influence between each other?

Responses: Thank you for your good question. The host-guest interactions between dye-load microenvironment catalyst \mathbf{H}_F and substrates 9-nitroanthracene (1:2), 1-nitropyrene (1:1), and 4-hydroxynitrobenzene (1:2) were observed in the microcalorimetric titration experiments. Then we carried out fluorescence titration experiments to explain the host-guest chemistry between the nitro substrates and dye-loaded host \mathbf{H}_F , which were further compared with the host-guest chemistry between nitro substrates and empty host \mathbf{H}_1 . The Job's plot fitting and Hill plot fitting of these fluorescence titration indicated that both the dye-loaded host \mathbf{H}_F and the empty host \mathbf{H}_1 exhibited the same inclusion number and comparable associate constants with the corresponding substrates, we proposed that the presence of fluorescein in the microenvironment did not restrict the encapsulation of substrates for further photocatalytic conversion.

Docking calculations to confirm the theoretically optimization of the clathrate structure of \mathbf{H}_F indicated that the encapsulated \mathbf{FI} molecule is stuck at one of the openings of \mathbf{H}_1 and it occupies only a little interior cavity, providing possibility for further encapsulating of nitro substrates. Meanwhile, the addition of nitro substrates into the \mathbf{H}_F solution could not trigger

the emission recovery of fluorescein, indicating that the encapsulation of nitro substrates could not squeeze out the **FI**. We have added the MS data of the **H_F** upon addition of one of the model substrate 4-nitrophenol (**S**), which exhibited the species of [**H_F**·2**S**·PF₆]⁵⁺ at $m/z = 1208.2942$, showing a 1:2 binding ratio between **H_F** and **S** that coincides with the results of ITC experiment (Supplementary Fig. 5). These results all suggested that the encapsulation of nitro substrates in the dye-loaded microenvironment catalyst did not squeeze the fluorescein out of the capsule, but formed the co-encapsulation clathrate.

Q5: Along this line, how about the influence from electron donor agents like N,N'-dimethylaniline, triethylamine, 1-phenylpyrrolidine, N,N'-diethylaniline or tripropylamine? The crystal structure shows many disordered small molecules inside the cage, so is there any evidence that these molecules cannot enter the cage cavity? This is an important issue because they are considered to determine “electron injection kinetics”.

Responses: Thank you for your good question. To investigate the potential interactions between the electron donors and the microenvironment catalyst **H_F**, fluorescence titration of the **H_F** upon addition of electron donors were displayed. With addition of the same proportion according to the photocatalytic reaction, significant luminescence quenching was observed. We proposed that the interactions between the electron donor and **H_F** was essential to trigger the quenching and affect the electron donating kinetics of the photoreduction conversion. We recognized that N, N'-dimethylaniline was the strongest quencher, tripropylamine was the weakest one (Supplementary Fig. 29a), obeying the same tendency to that of the yield ratio between the azo and amino product. As the fluorescein was encapsulated in the capsule and was strongly shielded by the microenvironment, the electron donors should be entered into the microenvironment to interact with the **FI** and injected the electrons.

Q6: Similar concern is about how to practically control “modifying electron injection kinetics of a microenvironment catalyst”. The diffusion coefficients of the mentioned electron donors are only slightly different (SI Table 3). It is also confusing in discussion like “As a larger molecular size of electron donors always results in slower entry kinetics or electron injection kinetics into the microenvironment of **H_F** through the openings, a smaller estimated diameter (**D**) with faster electron injection kinetics of the electron donor should allow the rapid and continuous reduction of the substrate and intermediates”. How the size of these donor molecules impact the entry kinetics or electron injection kinetics, through ingress of donor molecules into cage or other reason? If the kinetics is related to the exchange rate of the substrates, reduction actually happens outside the cage in solution.

Responses: Thank you for your good question. From the view of mechanism, after the intramolecular PET process, the electron donors need to approach the oxidized **FI** to donating electrons for another cycle of reduction. Due to the shielded cavity and openings of **H_F**, the hydrodynamic radius of the electroneutral electron donors was chosen as the factor to indicate the kinetics of entering the host cavity. Upon the addition of the same proportion electron donors according to the photocatalytic reaction to the same **H_F** solution, our experimental results revealed that N, N'-dimethylaniline was the strongest quencher, while tripropylamine

was the weakest one (Supplementary Fig. 29a). The positive relationship of the yield ratio between the azo and amino products with the estimated diameter (shown as $\pi d^2/4$) and the emission quenching efficiency of the corresponding electron donors for the photoreduction (Fig. 5b) further support our postulation and verify the presence of interactions between the electron donors and the microenvironment catalyst.

Meanwhile, in the case of 2,4,6-triphenylnitrobenzene, which has a molecular size that is larger than the opening of H_F , was used in Na_2S system, no amino product was detected under standard conditions. In the case of 1-nitropyrene was used in the tripropylamine system, only amino product could be detected. Meanwhile, the size-dependent photoreduction process and the inhibition competing experiments indicated the mechanism of enzymatic reaction that the photo-reduction occurred inside the cavity of H_F .

Furthermore, as H_F only encapsulated one 1-nitropyrene molecule and 1-aminopyrene was the only photoreduction product. The initial conversion rate 1-aminopyrene with different electron donors were used to reveal the electron injection rates of the correspond electron donor. As shown in Supplementary Fig. 29b, the positive correlation between the initial rate of the conversion and the estimated diameter of electron donors demonstrated that the entrance kinetics of electron donors into H_F dominated the electron injection kinetics and the photoreduction conversion rate.

Q7: Some typos need to be corrected, such as “Slective” should be changed to “Selective”.

Responses: Thank you for your good suggestion. We have carefully checked the whole manuscript and revised such mistakes in the revised manuscript.

Q8: Supporting information, page 71, there is an NMR spectrum showing different format from others.

Responses: Thank you for your good suggestion. We have updated the new NMR spectrum after purifying the product again.

Q9: The checkcif shows so many lever A/B alerts. The authors should add hydrogen for the disorder solvent molecules and solve or explain other level A/B alerts.

Responses: Thank you for your good suggestion. We have added hydrogen for the disorder solvent molecules. However, the still existed A alert in the checkcif file is caused by the weak diffraction intensity of the poor quality crystal, and the “short inter D...A contact” is caused by the presence of partly occupied solvent molecules with highly disordered atoms. We have added the explanation to the revised manuscript.

REVIEWER COMMENTS

Reviewer #1 (Remarks to the Author):

The authors have done an excellent job on revision following the suggestions from all the three reviewers. I recommend the publication of the current version as it is.

Reviewer #2 (Remarks to the Author):

The authors clearly responded to my and other referee's comments. The paper has been revised well and now can be accepted in the present form.

Reviewer #3 (Remarks to the Author):

As the host-guest chemistry in this work is complicated (including co-encapsulation of F1, different substrates and electron donor amines) but essential to the mechanism explained by authors, more evidenced proofs are needed to clarify queries from Reviewers #2 and 3#.

A suggestion for authors is to carried out the following experiments: 1) NMR titrations of H1 + F1, H1 + substrate, H1 + substrate-1 + substrate-2, H1 + substrate + amine; 2) DOSY NMR of H1 + F1, H1 + substrate, H1 + substrate-1 + substrate-2, H1 + substrate + amine.

NMR titration experiment between H1 and F1 in Fig. S16 just shows very slight shift of Ha and some protons on F1 without appearance of two sets of signals corresponding to free F1 and hosted F1, which is typical of fast exchange of the guest but not strong binding of guest. Author believe that at 273 K "the peaks of F1 and H1 were all broadened and the signal of Ha downfield shifted more significant, which indicated the existing of the slow exchange of guests" in Fig. S17 than at 298 K, therefore, the NMR titration and DOSY experiments should be performed at 273 K to testify this conclusion.

Although authors have performed microcalorimetric titrations and luminescence titrations, these experiments may just indicate there are interactions between cages and guests, no matter inside or outside. The simulation of H1+F1 with the result that F1 is stuck at one of windows of H1 may be possible, unfortunately, whether F1 can be replaced by substrates (100 equiv.) in catalytic process is not evidenced anyhow.

Reviewer: 3**Reviewer's Comments:**

As the host-guest chemistry in this work is complicated (including co-encapsulation of **FI**, different substrates and electron donor amines) but essential to the mechanism explained by authors, more evidenced proofs are needed to clarify queries from Reviewers #2 and 3#.

A suggestion for authors is to carried out the following experiments: 1) NMR titrations of **H₁** + **FI**, **H₁** + substrate, **H₁** + substrate-1 + substrate-2, **H₁** + substrate + amine; 2) DOSY NMR of **H₁** + **FI**, **H₁** + substrate, **H₁** + substrate-1 + substrate-2, **H₁** + substrate + amine.

NMR titration experiment between **H₁** and **FI** in Fig. S16 just shows very slight shift of Ha and some protons on **FI** without appearance of two sets of signals corresponding to free **FI** and hosted **FI**, which is typical of fast exchange of the guest but not strong binding of guest. Author believe that at 273 K “the peaks of **FI** and **H₁** were all broadened and the signal of Ha downfield shifted more significant, which indicated the existing of the slow exchange of guests” in Fig. S17 than at 298 K, therefore, the NMR titration and DOSY experiments should be performed at 273 K to testify this conclusion.

Although authors have performed microcalorimetric titrations and luminescence titrations, these experiments may just indicate there are interactions between cages and guests, no matter inside or outside. The simulation of **H₁**+**FI** with the result that **FI** is stuck at one of windows of **H₁** may be possible, unfortunately, whether **FI** can be replaced by substrates (100 equiv.) in catalytic process is not evidenced anyhow.

Responses: Thank you for your good suggestion. The titration experiments about **H₁** + **FI**, **H₁** + 4-nitrophenol, were well performed at 273 K and added in the supporting information as supplementary Figures 16 and 17, respectively. Significant down field shift of the OH protons and weak up field shifts of the aromatic protons were observed for the fluorescein and the substrate 4-nitrophenol, respectively.

The titration experiments about **H₁**+4-nitrophenol+9-nitroanthracene, **H₁**+ 4-nitrophenol + N,N-diethylaniline at 273 K were listed in supplementary Figures 19 and 20, respectively. The identical chemical shifts of 4-nitrophenol protons in the mixtures to that of 4-nitrophenol itself in **H₁** solution demonstrated that substrate 9-nitroanthracene and amine N,N-diethylaniline could not squeeze the 4-nitrophenol out of the microenvironment. Wherein, the identical chemical shifts of 9-nitroanthracene protons in **H₁** solution with or without the substrate 4-nitrophenol suggested the co-encapsulation of the two substrates in one capsule.

DOSY experiments about **H₁** + **FI** (supplementary Figure 22), **H₁** + 4-nitrophenol (supplementary Figure 23) were performed at 273 K. The additional results supported that the encapsulation of the guest molecules in our catalytic system. We also noted that the signals of interactions in DOSY spectrum were commonly weak with much low resolution, even in two-component systems, partly due to the low proton ratio between the guest molecule and the host molecule containing 48 phenyl rings. No meaningful results were obtained in the complicated three-component systems.

The ^1H NMR titration experiment between \mathbf{H}_1 and \mathbf{FI} was shown in the revised supporting information supplementary Figure 16. Significant down field shift of Ha signal in \mathbf{FI} molecule was observed (from 9.51 ppm to 9.86 ppm) with addition of 1.0 mM \mathbf{FI} , along with the up field shifts corresponding to aromatic protons, showing the interactions between \mathbf{H}_1 and \mathbf{FI} . The DOSY spectrum of the mixture of 1.0 mM \mathbf{H}_1 and 1.0 mM \mathbf{FI} showed that all signals of \mathbf{FI} and \mathbf{H}_1 had the same diffusion coefficient $D = 5.19 \times 10^{-10} \text{ m}^2\text{s}^{-1}$ (supplementary figure 22). The titration experiment about $\mathbf{H}_1 + \mathbf{FI} + 4\text{-nitrophenol}$ at 273 K was displayed (supplementary Figure 18). The down field shifts of OH proton signals in both \mathbf{FI} and 4-nitrophenol were observed simultaneously, which also matched well with the results in titration process separately and further proved the co-encapsulation of organic dye and the substrate in one capsule.

To verify the interactions of \mathbf{H}_1 and \mathbf{FI} even with addition of large amount of substrates. 100.0 mM nitrobenzene was added into the mixture of 1.0 mM \mathbf{H}_1 and 1.0 mM \mathbf{FI} in CD_3CN solution. As shown in supplementary Figure 21, the identical and unchanged up field shift of the Ha signal of \mathbf{FI} demonstrated \mathbf{FI} was encapsulated within the capsule \mathbf{H}_1 , it could not be replaced by the substrates in our catalytic process, even with 100 equiv. substrates.

REVIEWERS' COMMENTS

Reviewer #3 (Remarks to the Author):

The authors have provided experiments as suggested, although the results seem to be ambiguous.

Gaaino PDF Trial
www.gaaino.com

Reviewer's Comments:

Reviewer: 3

The authors have provided experiments as suggested, although the results seem to be ambiguous.

Responses: Thank you for your good suggestion. we have changed some words around the part of co-encapsulation of substrates and the part of reaction mechanism, and make conclusions more cautiously. Corresponding changes are marked in our revised manuscript.